# Innate immune signaling in trophoblast and decidua organoids defines differential antiviral defenses at the maternal-fetal interface

**Liheng Yang[1], Eleanor C Semmes[1,2], Cristian Ovies[1], Christina Megli[3,4], Sallie Permar[5], Jennifer B Gilner[6], Carolyn B Coyne[1,2]***

[1]Department of Molecular Genetics and Microbiology, Duke University School of Medicine, Durham, United States; [2]Duke Human Vaccine Institute, Duke University, Durham, United States; [3]Division of Maternal-Fetal Medicine, Division of Reproductive Infectious Disease, Department of Obstetrics, Gynecology and Reproductive Sciences, University of Pittsburgh Medical Center (UPMC), Pittsburgh, United States; [4]Magee Womens Research Institute, Pittsburgh, United States; [5]Department of Pediatrics, Weill Cornell Medical Center, Duke University Medical Center, Durham, United States; [6]Division of Maternal-Fetal Medicine, Department of Obstetrics and Gynecology, Duke University Medical Center, Durham, United States

**\*For correspondence:**
carolyn.coyne@duke.edu

**Competing interest:** The authors declare that no competing interests exist.

**Abstract** Infections at the maternal-fetal interface can directly harm the fetus and induce complications that adversely impact pregnancy outcomes. Innate immune signaling by both fetal-derived placental trophoblasts and the maternal decidua must provide antimicrobial defenses at this critical interface without compromising its integrity. Here, we developed matched trophoblast (TO) and decidua organoids (DO) from human placentas to define the relative contributions of these cells to antiviral defenses at the maternal-fetal interface. We demonstrate that TO and DO basally secrete distinct immunomodulatory factors, including the constitutive release of the antiviral type III interferon IFN-$\lambda$2 from TOs, and differentially respond to viral infections through the induction of organoid-specific factors. Finally, we define the differential susceptibility and innate immune signaling of TO and DO to human cytomegalovirus (HCMV) and develop a co-culture model of TO and DO which showed that trophoblast-derived factors protect decidual cells from HCMV infection. Our findings establish matched TO and DO as ex vivo models to study vertically transmitted infections and highlight differences in innate immune signaling by fetal-derived trophoblasts and the maternal decidua.

## Editor's evaluation

Yang et al., provide a scientifically sound and compelling manuscript characterizing mid-to-late gestation trophoblast and decidual organoids as ex vivo models to study vertically transmitted microbial infections, using human cytomegalovirus as a model pathogen. They demonstrate organoids have tissue-specific immunological responses and susceptibilities to viral infection.

# Introduction

The maternal-fetal interface is comprised of both the maternal-derived decidua and the fetal-derived placenta, which together support fetal growth, maintain maternal-fetal immunotolerance, and provide immunologic defenses to protect the fetus from infections. The relative contributions of maternal- and fetal-derived cells to these critical functions remain largely undefined given the limitations of models that recapitulate the complex nature of the human maternal-fetal interface.

The fetal-derived placenta contains distinct trophoblast subsets throughout gestation. Trophoblast stem cells located in chorionic villi, the branch-like projections that directly contact maternal blood, give rise to cytotrophoblasts (CTBs), the proliferative mononuclear cells of the placenta, the syncytiotrophoblast (STB), a multinucleated contiguous cell layer that covers the villous surfaces, and extravillous trophoblasts (EVTs). EVTs remodel the maternal microvasculature to form spiral arteries, which ultimately deliver maternal blood to the surface of chorionic villi. On the maternal side, decidualization of the endometrium is required for implantation and appropriate placentation. Taken together, the decidua and placenta maintain immunotolerance throughout gestation and provide the fetus with essential hormones and nutrients to support its growth.

While the placenta and decidua play complementary roles in fetal development, their contributions to antimicrobial defenses are less defined (reviewed in *Megli et al., 2021*; *Semmes and Coyne, 2022*). The placenta forms a powerful barrier to infections, yet certain congenital pathogens (e.g., *Toxoplasma gondii*, Zika virus (ZIKV), HIV, Rubella, and human cytomegalovirus [HCMV]) can be vertically transmitted to infect the fetus and in some cases, cause disease and/or pregnancy loss. Studies of tissue explants and histopathologic analysis of the placenta suggest that the decidua and chorionic villi have differing roles in the pathogenesis of vertically transmitted infections. The decidua may be a primary site of replication for several microorganisms associated with congenital disease, including HCMV (*McDonagh et al., 2006*; *Pereira et al., 2003*; *Weisblum et al., 2015*), ZIKV (*Guzeloglu-Kayisli et al., 2020*; *Platt et al., 2018*; *Weisblum et al., 2017*), and *Listeria monocytogenes* (*Rizzuto et al., 2017*), suggesting it may serve as a reservoir for these infections. By contrast, the placenta is largely resistant to infections and possesses intrinsic mechanisms of innate immune defense. For example, human trophoblasts constitutively release antiviral interferons (IFNs) that restrict infection in both autocrine and paracrine manners (*Bayer et al., 2016*; *Corry et al., 2017*). These IFNs are of the type III class, which include IFNs-$\lambda$ 1–3 in humans (reviewed in *Lazear et al., 2019*; *Wells and Coyne, 2018*). Constitutive release of IFNs is a unique feature of trophoblasts as IFNs are tightly regulated in most cell types and induced only after detection of the products of viral replication (reviewed in *Lazear et al., 2019*). Taken together, these findings suggest that maternal- and fetal-derived tissue differentially respond to and control viral infections; however, the lack of available models to fully interrogate the differences between maternal- and fetal-derived cells have limited a complete understanding of their respective roles in antiviral defense.

The paucity of tractable models to study the human maternal-fetal interface and the limitation of accessing this interface in vivo without disrupting pregnancy has limited our understanding of the pathogenesis of and immune responses against many congenital pathogens, including HCMV. HCMV is a species-specific β-herpesvirus that can only infect humans, and animal models using guinea pig (GPCMV) or rhesus macaque (RhCMV) cytomegaloviruses rely on viruses that are genetically disparate from HCMV and do not fully recapitulate the human maternal-fetal interface (*Roark et al., 2020*). Prior studies on placental HCMV infection have used trophoblast progenitor cells, EVT cell lines, and primary trophoblasts; however, these cell lines and isolated primary cells do not model the cellular complexity of the human placenta (*Njue et al., 2020*). Placental and decidual explants have also been employed to model HCMV infection, yet these human explants have high inter-individual variability, which hinders reproducibility, moreover, there is a limited window of viability in cultured ex vivo tissue (*Fisher et al., 2000*; *Weisblum et al., 2017*; *Weisblum et al., 2015*). Therefore, novel models of the human maternal-fetal interface are needed to understand the contributions of maternal decidual cells and fetal-derived placental cells to immune defense against and pathogenesis of placental HCMV infection.

The recent development of trophoblast (TO) and decidua organoids (DO) from first trimester placental tissue has greatly expanded the models available to study the human maternal-fetal interface (*Turco et al., 2017a*; *Turco et al., 2018*). Previous work has shown that isolation and propagation of trophoblast stem cells from first trimester placental chorionic villi lead to the development of

three-dimensional organoids that differentiate to contain all lineages of trophoblast cells present in the human placenta and release well-known pregnancy hormones such as human chorionic gonadotropin (hCG) (*Turco et al., 2018*). Two-dimensional and organoid-based models of first trimester stem cells derived from villous tissue have also been reported and recapitulate key aspects of trophoblast biology (*Karvas et al., 2022*; *Okae et al., 2018*; *Shannon et al., 2017*; *Sheridan et al., 2021*). Similarly, DOs isolated from uterine glands recapitulate the transcriptional profile of their tissue of origin and respond to hormonal cues (*Turco et al., 2017a*). These organoid models are ideal to interrogate differences between maternal- and fetal-derived cells, but matched DO and TOs have not yet been used to model congenital infections.

To directly compare antiviral signaling pathways between the human placenta and decidua, we generated TO and DO from matched mid-to-late gestation (26–41 weeks of gestation) human placentas and profiled their release of cytokines, chemokines, and other factors at baseline and under virally infected conditions. These studies revealed that TOs basally secrete IFN-$\lambda$2, which does not occur in matched DOs. To determine whether maternal- and fetal-derived cells respond differently to viral infections, we compared the transcriptional profiles and immunological secretomes from organoids treated with a ligand to stimulate toll-like receptor 3 (TLR3) signaling. TO and DO differentially responded to TLR3 activation through the secretion of organoid-specific cytokines and chemokines. Additionally, we defined the differential susceptibility of trophoblast and decidua organoids to HCMV infection and identified key differences in the innate immune responses of these organoids to HCMV infection. Finally, to simulate the communication between trophoblasts and the decidua epithelium in vivo, we established a co-culture system between matched TO and DO, which showed that fetal-derived trophoblasts protect the decidual epithelium from HCMV infection. Taken together, these studies highlight the differential responses of decidual and placental cells at the maternal-fetal interface to viral infections and provide new models to understand innate immune responses against placental infections.

## Results

### Establishment and characterization of human trophoblast and decidual organoids from mid-to-late gestation tissue

To generate matched human TOs and decidual organoids (DOs), we used human placental tissue isolated from the second and third trimesters of human pregnancy (26–41 weeks gestation) and cultured progenitor cells isolated from chorionic villi or uterine glands of the decidua (*Figure 1A*). Given that previous work developed TOs utilizing tissue isolated from the first trimester (*Haider et al., 2018*; *Marinić et al., 2020*; *Okae et al., 2018*; *Sheridan et al., 2021*; *Turco et al., 2017a*; *Turco et al., 2018*), we optimized the isolation and propagation procedures for isolation of cells from mid-to-late gestation chorionic villi. This optimization included extended mechanical disruption of tissue and the addition of supplements including nicotinamide to promote progenitor cell differentiation. Although the abundance of trophoblast stem/progenitor cells decreases after the first trimester (*Turco et al., 2018*), we successfully isolated progenitor cells and generated organoids from all placental tissue harvested (15 unique placentas). Consistent with the presence of progenitor cells in full-term chorionic villi, we identified cells positive for the proliferation marker Ki67 in villi isolated from two late gestation placental villi used to generate TOs (*Figure 1—figure supplement 1A*). Isolated progenitor cells were embedded in Matrigel to allow for self-organization and propagated for ~2–3 weeks to generate organoid structures that were morphologically like those previously isolated from first trimester placentas (*Turco et al., 2017a*; *Turco et al., 2018*; *Figure 1B*, *Figure 1—figure supplement 1B*). Once established, organoids could be continuously expanded and passaged every 3–5 days for DOs and 5–7 days for TOs and were highly proliferative based on Ki67 immunostaining (*Figure 1—figure supplement 1C*). Confocal microscopy for cytoskeletal and adhesion markers including cytokeratin-19, actin, E-cadherin, and the Epithelial Cell Adhesion Molecule (EpCAM) confirmed the three-dimensional architecture of established organoids and verified that TOs and DOs expressed markers consistently with their tissues of origin (*Figure 1C and D* and *Figure 1—figure supplement 1D*). This also included the expression of the trophoblast-specific marker Sialic Acid Binding Ig Like Lectin 6 (SIGLEC6) in TOs but not DOs and the expression of the mucin MUC5AC in DOs but not TOs (*Figure 1C and D* and *Figure 1—figure supplement 1D*). TOs also expressed

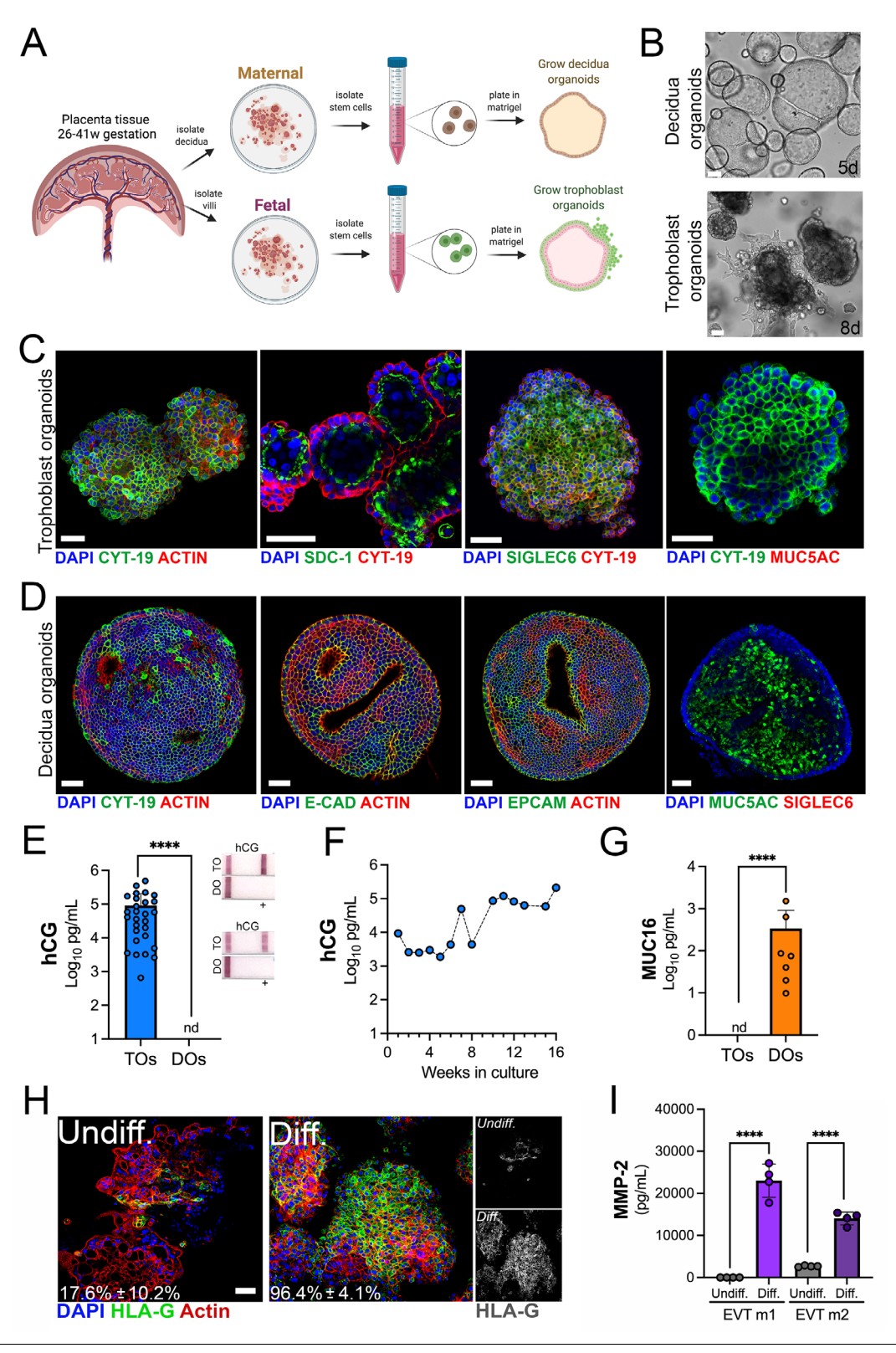

**Figure 1.** Long-term three-dimensional organoids cultures can be established from human mid- and late-gestation placental tissue. (**A**) Schematic representation of the derivation of trophoblast organoids (TOs) and decidual organoids (DOs) from mid-to-late gestation placental tissue. (**B**) Representative bright-field images of TOs and DOs after passaging in complete growth medium TOM and ExM, respectively at 5 days (DO, top) or 8 days

*Figure 1 continued on next page*

*Figure 1 continued*

(TO, bottom) post-passaging. Scale bar, 50 µm. (**C**) Confocal microscopy in TOs immunostained for (from left) cytokeratin-19, SDC-1, SIGLEC-6, cytokeratin-19 (in green) and actin, cytokeratin-19, or MUC5AC (in red). DAPI-stained nuclei are shown in blue. Individual channels are shown in *Figure 1—figure supplement 1D*. Scale bar, 50 µm. (**D**) Confocal microscopy in DOs immunostained for (from left) cytokeratin-19, E-cadherin (E-cad), EpCAM, or MUC5AC (in green) and actin or SIGLEC6 (in red). DAPI-stained nuclei are shown in blue. Individual channels are shown in *Figure 1—figure supplement 1D*. Scale bar, 50 µm. (**E**) Levels of hCG in conditioned medium (CM) isolated from TOs (blue) or DOs (orange) as determined by Luminex. At right, over the counter (OTC) pregnancy tests for hCG in two matched TO and DO lines. (**F**) Levels of hCG in CM isolated from TOs throughout a 16-week culture period as determined by Luminex. (**G**) Levels of secreted Mucin-16 (MUC16) in CM isolated from TOs (light blue) or DOs (orange) as determined by Luminex assays. (**H**) Confocal microscopy in TOs differentiated into an EVT-enriched phenotype (Diff., right) or cultured under standard growth conditions (Undiff., left), and immunostained with HLA-G (in green). Actin is shown in red and DAPI-stained nuclei in blue. At right, black and white images of HLA-G in undifferentiated (top) or differentiated (bottom) conditions. (**I**) Levels of MMP-2 in the media of undifferentiated (undiff.) or EVT-differentiated (diff.) TOs as assessed by Luminex assays. Shown are media isolated from both stages of EVT differentiation, medium 1 (EVT m1), which contains NRG1, and medium 2 (EVT m2), without NRG1. In (**E**), (**G**), and (**I**), each symbol represents an individual CM preparation and significance was determined by Mann-Whitney U-test. ****, p<0.0001 and nd, not detected. EVT, extravillous trophoblast; hCG, human chorionic gonadotropin.

The online version of this article includes the following video and figure supplement(s) for figure 1:

**Figure supplement 1.** Characterization of trophoblast (TO) and decidua organoids (DO).

**Figure 1—video 1.** Three-dimensional image reconstruction of SDC-1 immunostaining in trophoblast organoids (TOs).

https://elifesciences.org/articles/79794/figures#fig1video1

high levels of Syndecan-1 (SDC-1), a specific marker of the fused STB (*Figure 1C*). The localization of SDC-1 in TOs shows that like organoids derived from first trimester placental tissue (*Turco et al., 2018*), the apical surface of the STB is inward facing (*Figure 1—video 1*). We also found that TOs, but not DOs, secreted the trophoblast-specific hormone hCG as assessed by Luminex assays and over the counter (OTC) test strips (*Figure 1E*). The levels of hCG were stable in TOs propagated for extended periods of time (*Figure 1F*). In addition, we confirmed that DOs released high levels of the secreted mucin MUC16 as determined by Luminex assays, which was not present in TOs (*Figure 1G*).

Trophoblast stem cells and trophoblast-derived organoids derived from the first trimester placental tissue can be differentiated to an EVT-enriched phenotype through a two-step process that includes treatment with Neuregulin-1 (NRG1) (*Okae et al., 2018*; *Sheridan et al., 2021*). Similarly, we found that this differentiation process significantly increased the presence of HLA-G positive cells in TOs isolated from full-term tissue (from ~18% positive cells to ~96% positive cells) (*Figure 1H*) and were morphologically like those described previously (*Figure 1—figure supplement 1E*; *Sheridan et al., 2021*). Consistent with an enrichment of HLA-G positive EVTs, we detected elevated levels of matrix metalloproteinase-2 (MMP-2), which is expressed at high levels in first trimester EVTs (*Isaka et al., 2003*; *Shimonovitz et al., 1994*), in media collected from EVT differentiated TOs (*Figure 1I*).

Collectively, these data show that TOs and DOs are morphologically and functionally distinct and that they recapitulate human placental trophoblasts and decidua glandular epithelial cells.

## Distinct transcriptional profiles in TOs and DOs

Next, we performed transcriptional profiling of established TO and DO lines using bulk RNASeq. We found that TOs and DOs were transcriptionally distinct based on differential expression analysis using DESeq2 followed by principal component analysis (PCA) (*Figure 2A*, *Figure 2—figure supplement 1A B, C*). TOs highly expressed transcripts specifically enriched in the placenta, including HSD3B1, XAGE2 and 3, GATA3, and ERVW-1, which were absent or expressed at very low levels in DOs (*Figure 2B*, *Figure 2—figure supplement 1C and D, E*). In addition, TOs highly expressed transcripts for PSGs, ERVW-1, and CGA, which were not expressed in DOs (*Figure 2B*). The levels of expression of these transcripts were comparable to those in primary human trophoblasts (PHTs), confirming that TOs recapitulate the transcriptional signature of primary trophoblasts (*Figure 2B*). These markers were consistent with the presence of the STB (e.g., CGA, PSGs, and HSD3B1), CTBs (e.g., TP63), and EVTs (e.g., ITGA5) (*Figure 2B*). In contrast, DOs expressed markers associated with the decidua and

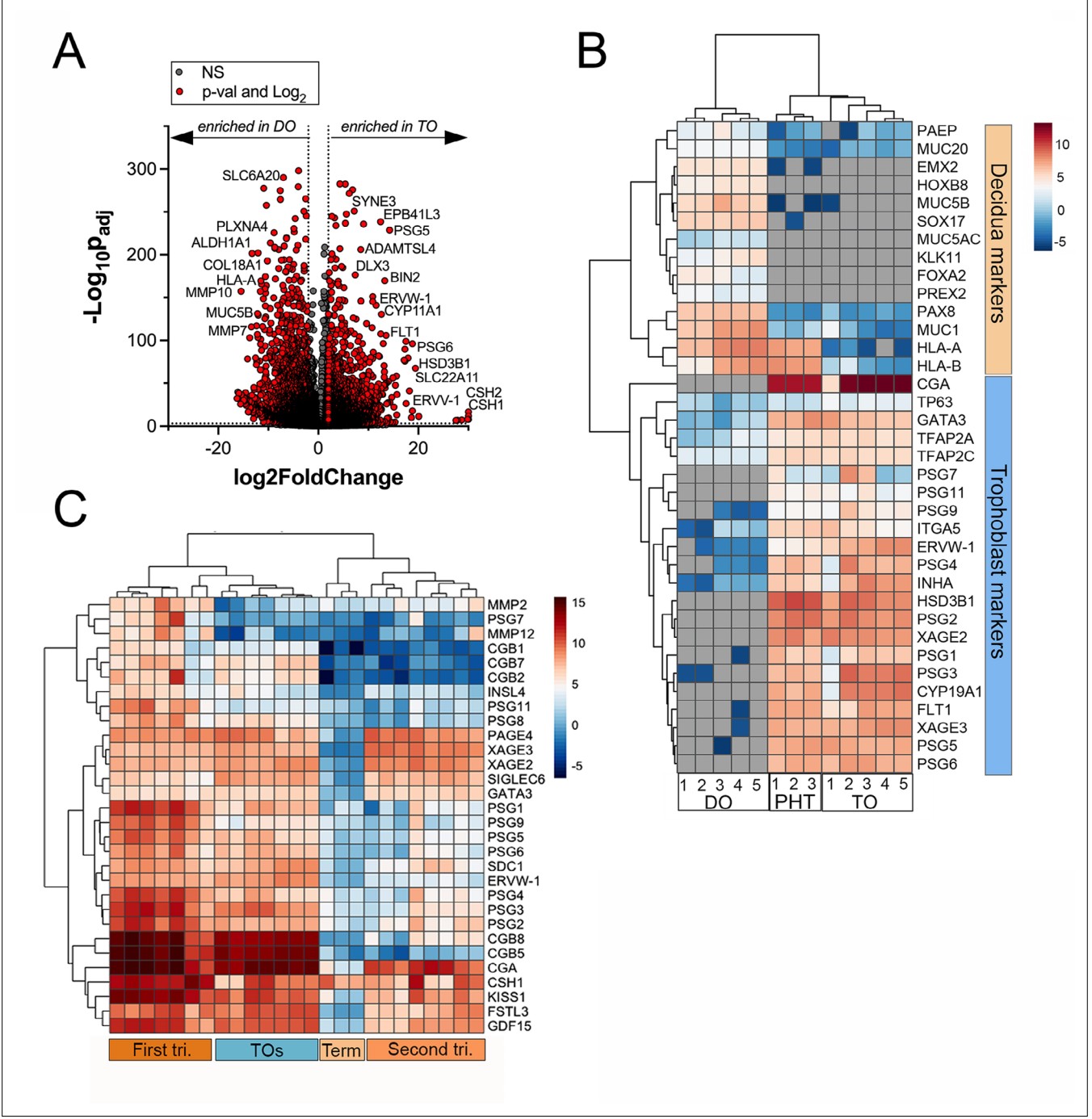

**Figure 2.** Whole genome transcriptional profiling of trophoblast (TO) and decidua organoids (DO). (**A**) Volcano plot demonstrating differentially expressed transcripts in TOs or DOs as assessed by DeSeq2 analysis. Gray circles represent genes whose expression was not significantly changed and red denotes transcripts significantly enriched in TOs (right) or DOs (left) (significance was set at $p<0.01$ and $\log_2$ fold-change of $\pm2$). (**B**) Heatmap (based on $\log_2$ RPKM values) of transcripts expressed in DOs, TOs, or primary human trophoblasts (PHTs) cells that are associated with markers of trophoblasts (top, blue) or the decidua (bottom, orange). Key at right. Red indicates high levels of expression, blue indicates low levels of expression, and gray indicates no reads detected. Hierarchical clustering is shown on left and top. (**C**) Heatmap (based on $\log_2$ RPKM values) of transcripts expressed in TOs or human placental tissue isolated from the first or second trimesters, or from full-term tissue. Key at right. Red indicates high levels of expression, blue indicates low levels of expression, and gray indicates no reads detected. Hierarchical clustering is shown on left and top.

The online version of this article includes the following figure supplement(s) for figure 2:

**Figure supplement 1.** RNASeq of TOs and DOs.

decidual stromal cells, including MUC1, MUC5A, MUC5B, SOX17, and HOXB8 which were absent or expressed at low levels in TOs and PHT cells (*Figure 2B*, *Figure 2—figure supplement 1F and G*). To further confirm that TOs and DOs recapitulated their tissues of origin, we profiled the expression of three members of the chromosome 19 microRNA family (C19MC), which are specifically expressed in the human placenta (*Mouillet et al., 2015*). We found that miR-516B-5p, miR-517A-3p, and miR-525-5p were expressed at high levels in TOs, comparable to levels observed in freshly isolated chorionic villi, but were not expressed in DOs (*Figure 2—figure supplement 1I*). Finally, we found that TOs isolated from placentas obtained from male fetuses expressed Y-linked genes, which were absent in female-only derived DOs (*Figure 2—figure supplement 1H*).

To determine if TOs derived from full-term tissue recapitulated the transcriptional signature of placental tissue throughout gestation, we performed comparative transcriptional profiling of TOs and publicly available RNASeq data sets from placental tissue isolated from the first and second trimesters, or from full-term tissue. We performed differential expression analysis of these data sets and compared the expression profiles of key pregnancy hormones and other factors associated with placental structure and function. Hierarchical clustering showed that TOs most closely clustered with samples derived from first trimester placental tissue (*Figure 2C*). In addition, TOs expressed high levels of pregnancy hormones such as CGA and various CGBs similar to first trimester tissue and also expressed factors including kisspeptin-10 (KISS1), which is expressed at higher levels in first trimester tissue compared to full-term tissue (*Bilban et al., 2004*; *Figure 2C*).

Collectively, these data establish matched TOs and DOs as model systems of the maternal-fetal interface and demonstrate that they can be derived from tissue after the first trimester, including from mid-to-late gestation. Furthermore, these data suggest that TOs derived from full-term tissue recapitulate the expression profile of first trimester placental tissue.

## Comparative immune secretome profiling of TOs and DOs

We have shown previously that human chorionic villous explants release numerous cytokines, chemokines, and growth factors that impact both maternal and fetal antimicrobial defenses (*Corry et al., 2017*; *Megli et al., 2021*). Similarly, we have shown that PHTs isolated from mid-gestation and full-term placentas secrete some of these same immunomodulatory factors (*Corry et al., 2017*; *Megli et al., 2021*). We therefore profiled the immunological secretome of matched TOs and DOs isolated from independent placentas to define basal differences in the secretome of cells that comprise the maternal-fetal interface. To do this, we used multianalyte Luminex-based profiling of 105 cytokines, chemokines, growth, and other factors from conditioned media (CM) isolated under basal unstimulated conditions. Factors whose secretion was reliably measured over 50 pg/mL were defined as present in CM. We found that TOs secreted only two cytokines (IL-6 and IL-28A/IFN-$\lambda$2) and three immune-regulated secreted soluble receptors (sTNF-R1, gp130/sIL-6R$\beta$, and sTNF-R2) at baseline (*Figure 3A*). By contrast, DOs released a larger number of immune factors overall, with 21 detected in DO-derived CM at baseline (*Figure 3B*). PCA of these Luminex profiles identified factors contributing to the differential nature of the secretome between TOs and DOs, which included the enrichment of factors including CXCL1, CXCL5, and IL-8 in DOs versus IL-6 and IL28A/IFN-$\lambda$2 in TOs (*Figure 3C–F*). Taken together, these data define the unique immunologic secretome of key cells that comprise the maternal-fetal interface and show that tissue-derived organoids recapitulate some of the secretory phenotypes observed in tissue explants and/or primary cells.

## TOs constitutively secrete antiviral IFN-$\lambda$2

We showed previously that PHT cells isolated from full-term placentas and mid-gestation chorionic villi constitutively release type III IFNs, which protect trophoblasts from viral infections (*Bayer et al., 2016*; *Corry et al., 2017*). We found that TOs recapitulated this phenotype and released high levels of IFN-$\lambda$2, but not type I IFNs or IFN-$\lambda$1 (*Figure 4A*). In contrast, DOs did not basally secrete any detectable IFNs (*Figure 4B*). The levels of released IFN-$\lambda$2 were comparable to those from PHT cells and from matched cultured villous explants (*Figure 4C*). We monitored the release of IFN-$\lambda$2 over the growth period of TOs and found that its release was detectable by ~10 days post-plating and reached high levels by >14 days in culture, which correlated with similar trends for IL-6 (*Figure 3—figure supplement 1*). We confirmed that CM isolated from TOs, but not DOs, induced interferon-s (ISGs) in cells treated with this CM using a HEK293-based reporter assay

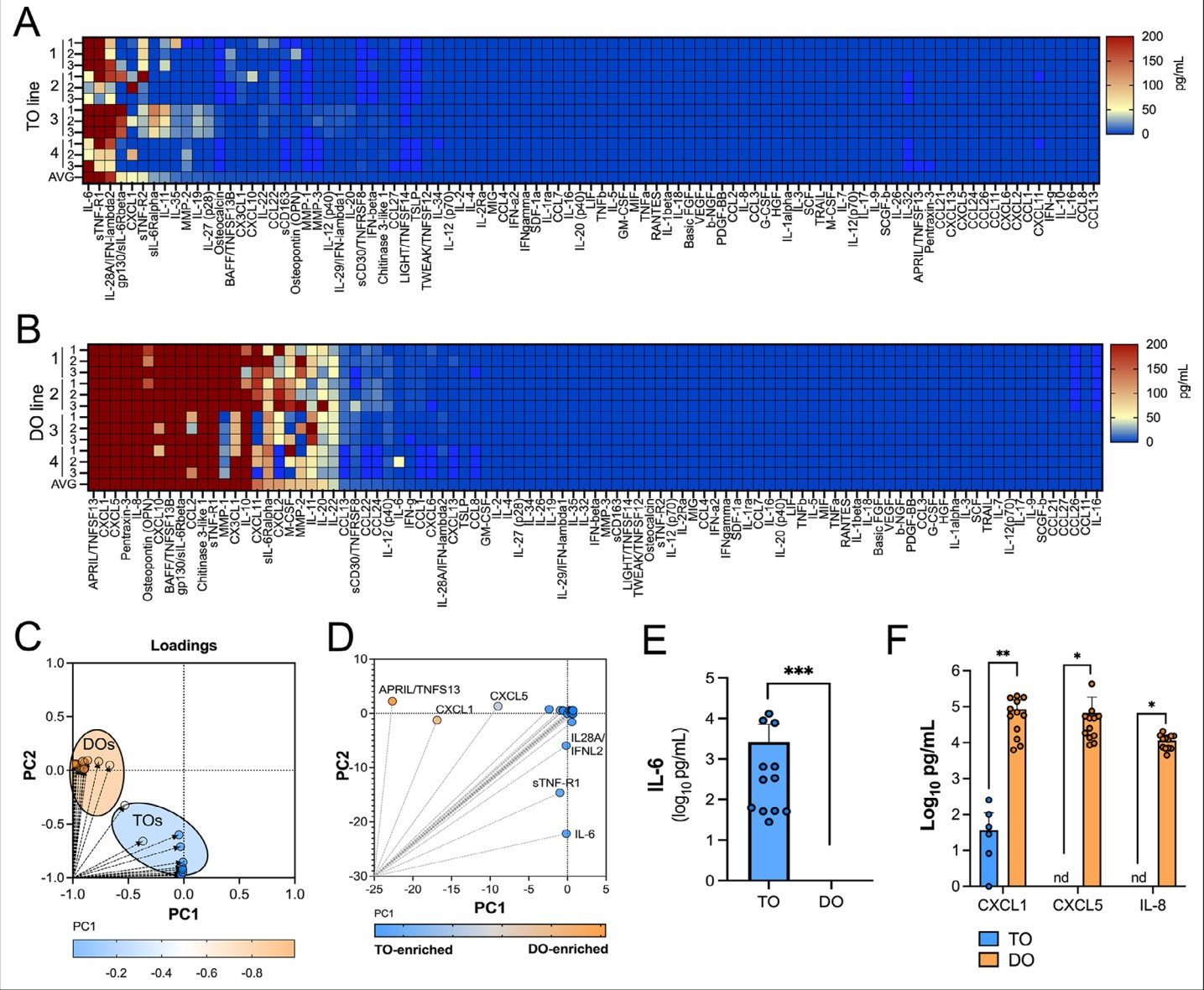

**Figure 3.** Basal cytokine and chemokine secretion profiles from matched trophoblast (TO) and decidua organoids (DO). (**A, B**) Heatmap of 12 conditioned medium (CM) preparations generated from four established TO (**A**) and DO (**B**) lines analyzed by Luminex-based multianalyte profiling for the indicated cytokines, chemokines, and growth factors (at bottom). Scale is shown at right. (**C, D**) Principal component analysis of Luminex data shown in (**A**) and (**B**) indicates that the basal secretion between TOs and DOs is unique (**C**) and identifies the factors that contribute to these differences (**D**). (**E**) Basal secretion of IL-6 in TOs, but not DOs, as assessed by Luminex. (**F**) Basal secretion of CXCL1, CXCL5, and IL-8 in CM from TOs (in blue) or DOs (in orange). In (**E**) and (**F**), each symbol represents an individual CM preparation and significance was determined by Mann-Whitney U-test (**E**) or two-way ANOVA with Šídák's multiple comparisons tests (**F**). *** $p < 0.001$, ** $p < 0.01$, * $p < 0.05$.

The online version of this article includes the following figure supplement(s) for figure 3:

**Figure supplement 1.** IL-6 and IFN-λ 2 release in TOs over time.

(*Figure 4D*). Consistent with this, CM isolated from TOs was antiviral and reduced ZIKV infection in CM-treated non-placental cells including U2OS cells (*Figure 4E*) and in DOs (*Figure 4F and G*), confirming that TO-derived CM exerts potent antiviral effects on non-placental cells. Finally, we found that TOs differentiated into an EVT-enriched phenotype exhibited significantly reduced levels of secreted IFN-λ 2, suggesting that EVTs are not the primary producers of IFN-λ 2 in TOs (*Figure 4H*).

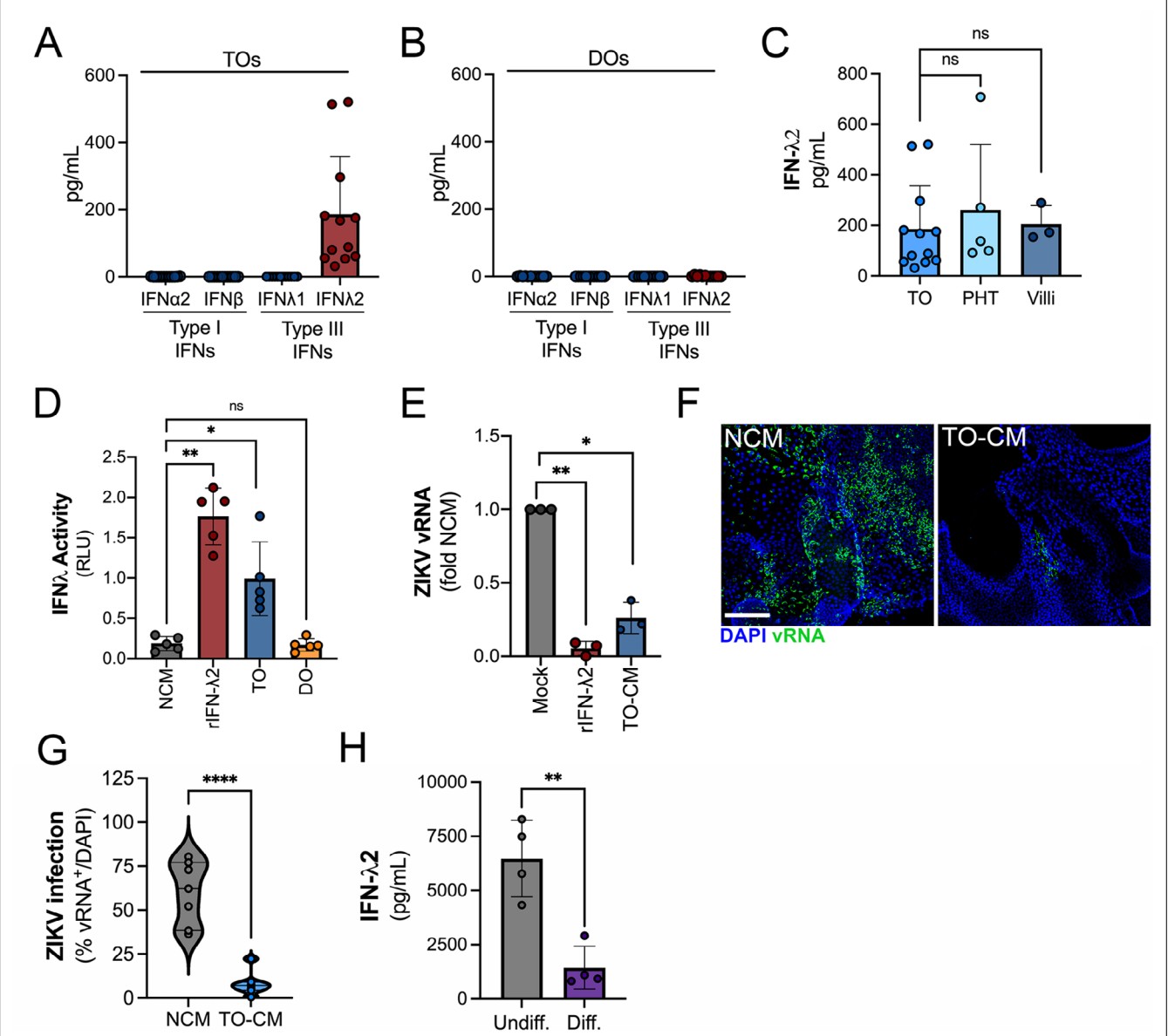

**Figure 4.** Trophoblast organoids secrete high levels of antiviral IFN-λ 2. (**A, B**) Comparison of the levels of the type I IFNs IFN-α2 and IFN-β and type III IFNs IFN-λ 1 and IFN-λ 2 in conditioned medium (CM) isolated from TOs (**A**) or DOs (**B**) as determined by Luminex assays. (**C**) Comparison of the levels of IFN-λ 2 in CM isolated from TOs, matched villous tissue, or in primary human trophoblasts (PHTs). (**D**) IFN-λ activity shown as relative light units (RLUs) of CM isolated from TOs (blue) and DOs (orange) as assessed using an IFN-λ reporter 293T cells line. NCM (non-conditioned medium, gray) was as negative control, NCM supplemented with recombinant human IFN-λ 2 (100 ng) was used as a positive control (red). (**E**) Human osteosarcoma U2OS cells were treated with CM isolated from TOs (blue) or with recombinant IFN-λ 2 (100 ng) for 24 hr and then infected with ZIKV for 24 hr (in the presence of CM). Level of ZIKV replication was assessed by viral RNA as determined by qPCR. Data are shown as a fold change in infection from NCM-treated cells. (**F**) Representative confocal micrographs of DOs treated with NCM or CM isolated from TOs for 24 hr, then infected with ZIKV for 72 hr. Shown in green is viral double stranded RNA, a replication intermediate. DAPI-stained nuclei are shown in blue. Scale bar, 50 µm. (**G**) Automated image analysis of ZIKV infection in ZIKV-infected DOs treated with NCM or TO-derived CM, as described in (**F**). Imaged were captured from three different experiments. (**H**) Comparison of the levels of IFN-λ 2 in CM isolated from matched TOs grown under conditions that promote EVT differentiation (Diff., right) or under standard culture conditions (Undiff., left). Data in all panels are mean ± standard deviation from at least three independent CM preparations. Significance was determined by a Student's t-test, Mann-Whitney U-test or by Kruskal-Wallis U-test with multiple comparisons. *p<0.05; **p<0.01; ****p<0.001, ns, not significant. In all, each symbol represents unique CM preparations. DO, decidua organoid; TO, trophoblast organoid.

## Differential induction of cytokine and chemokine networks in TOs and DOs in response to poly I:C treatment

Because we observed differences in the basal secretomes of TOs and DOs, we next determined whether TOs and DOs differentially respond to viral infections. To model RNA virus infection, we used poly (I:C), a synthetic ligand of TLR3, and profiled the transcriptional and secretory changes induced by poly I:C treatment of TOs and DOs. We found that both TOs and DOs robustly responded to poly I:C treatment, which corresponded with significant transcriptional changes as assessed by RNASeq (*Figure 5A and B*). Comparison of the transcripts induced by poly I:C treatment (as defined by significance of p<0.01 and $\log_2$ fold-change of ±2) of TOs and DOs revealed that TOs upregulated 448 transcripts whereas DOs upregulated 121 (*Figure 5—figure supplement 1A*). Of these, only 69 were shared between TOs and DOs, the majority of which (64 of 69) were ISGs (*Figure 5—figure supplement 1A and B*). Of the differentially induced transcripts, TOs induced the expression of chemokines including CXCL5 ($\log_2$ fold change 5.9), CX3CL1 ($\log_2$ fold change 7.1), and CCL22 ($\log_2$ fold change 7.6), which were the most induced cytokine or chemokine transcripts in poly I:C-treated TOs (*Figure 5C*). In contrast, DOs expressed chemokines at high basal levels, which were not further induced by poly I:C treatment and did not induce any CCL22 (*Figure 5C*). Transcripts most selectively induced in DOs included CD38 and the phospholipase PLA1A (*Figure 5C*).

To confirm the transcriptional changes described above, we performed parallel multianalyte Luminex profiling of 105 cytokines and chemokines in TOs and DOs treated with poly I:C. Both TOs and DOs mounted robust responses to poly I:C treatment that correlated with the release of high levels of select cytokines and chemokines (*Figure 5D*). Cytokines and chemokines with levels >5-fold above baseline were defined as induced by poly I:C treatment. The topmost induced factors in poly I:C-treated TOs were the chemokines CCL22, CXCL5, CXCL2, and CX3CL1 (*Figure 5D*, left), which was consistent with their transcriptional upregulation by RNASeq (*Figure 5C*). PCA of Luminex-based profiling of poly I:C treated TOs and DOs revealed organoid-specific responses to this treatment, which included the specific induction of CCL22 in poly I:C treated TOs (*Figure 5—figure supplement 1C*, *Figure 5E and F*). Factors induced by both TOs and DOs included the type III IFNs IFN-$\lambda$1 and IFN-$\lambda$2 (*Figure 5G and H*), which were the topmost induced factors in DOs. In contrast, the type I IFNs IFN-α2 and IFNβ were induced to significantly greater levels in TOs than DOs (*Figure 5—figure supplement 1D and E*). Other chemokines induced in TOs were basally expressed at high levels in DOs and were not further induced by poly I:C (*Figure 5—figure supplement 1F-J*).

## Differential susceptibility of TOs and DOs to HCMV infection

HCMV is the most common vertically transmitted pathogen and is associated with high rates of placental dysfunction, pregnancy loss, and congenital disease. However, how HCMV transmits across the placenta and how the maternal decidua and fetal-derived placenta respond to HCMV infection remains poorly defined. To address these questions, we infected TOs and DOs with HCMV strains AD169r (BADrUL131-Y4, tagged with GFP; *Wang and Shenk, 2005*) and TB40E (tagged with mCherry; *O'Connor and Shenk, 2011*). We found that DOs were highly permissive to HCMV infection as quantified by detectable GFP or mCherry expression of viral gene expression in most organoids (*Figure 6A, C and D*). In contrast, TOs were largely resistant to HCMV infection, with low levels of infection by AD169r and almost undetectable levels by TB40E, as assessed by fluorescence of tagged viral gene expression (*Figure 6B, C and D*). Consistent with this, analysis of the expression of key HCMV products using bulk RNASeq from infected TOs and DOs showed that the HCMV-associated factors RNA4.9, which is important for HCMV DNA replication (*Tai-Schmiedel et al., 2020*), and UL63 were significantly lower in TOs compared to DOs (*Figure 6E*).

Because we observed minimal, but detectable, HCMV infection in TOs, we next sought to determine whether the low levels of HCMV infection we observed occurred in EVTs, which have been proposed as targets of many teratogenic viruses including HCMV (reviewed in *Megli et al., 2021*). To do this, we infected TOs cultured under standard conditions or under conditions that induced their differentiation into EVTs. Despite the significant enrichment of HLA-G positive EVTs under differentiated conditions, we found that there was no significant enhancement of HCMV infection under these conditions (*Figure 6F and G*), suggesting that EVTs are not the primary targets of HCMV infection in TOs.

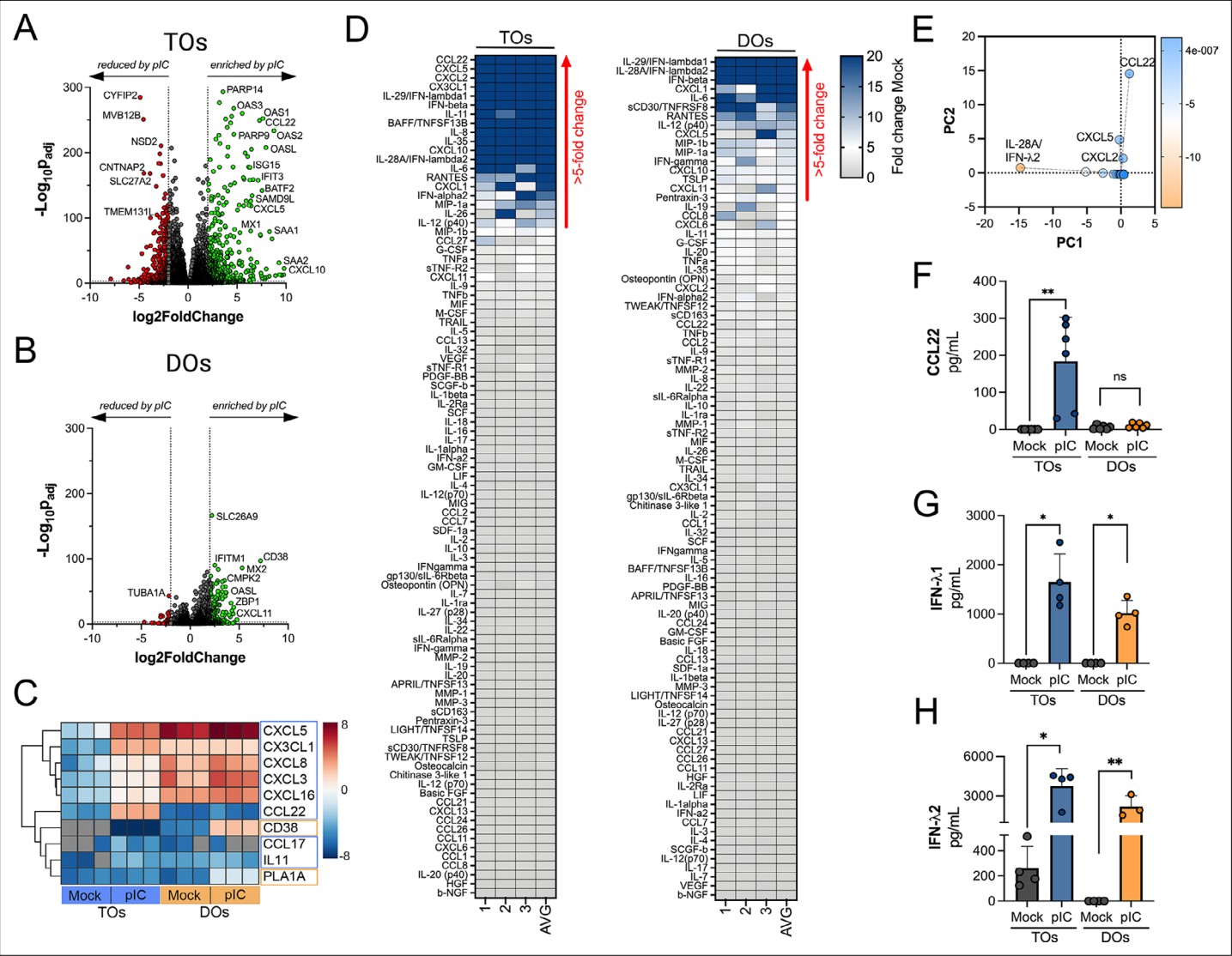

**Figure 5.** Differential cytokine and chemokine release in trophoblast organoids (TOs) and decidua organoids (DOs) treated with poly I:C. (**A, B**) Volcano plots demonstrating differentially expressed transcripts in TOs (**A**) or DOs (**B**) treated with poly I:C as assessed by DeSeq2 analysis. Gray circles represent genes whose expression was not significantly changed and green denotes transcripts significantly induced by poly I:C treatment whereas red denotes transcripts significantly reduced by poly I:C treatment. Significance was set at p<0.01 and log₂ fold-change of ±2. (**C**) Heatmap (based on log₂ RPKM values) of select transcripts induced uniquely by poly I:C treatment of TOs (blue boxes) or DOs (orange boxes). Key at right. Red indicates high levels of expression, blue indicates low levels of expression, and gray indicates no reads detected. Hierarchical clustering is on the left. (**D**) Heatmaps demonstrating the induction of factors at left (shown as fold change from mock-treated controls) from TOs (left panel) and DOs (right panel) treated with 10 μg poly (I:C) and analyzed by multiplex Luminex-based profiling for 105 cytokines, chemokines, and other factors. AVG denotes the average change in concentration of factors over conditioned medium (CM) isolated from three individual preparations. Dark blue denotes significantly induced factors compared with untreated controls. Gray or white denotes little to no change (scale at top right). The red arrow demonstrates factors with induction greater than fivefold change observed in the average of separate experiments. Data are from three individual CM preparations from at least three unique matched organoid lines. (**E**) Principal component analysis of Luminex data shown in left panels identify factors differentially induced by poly I:C treatment of TOs and DOs. (**F–H**) Levels of CCL22 (**F**), IFN-λ 1 (**G**), and IFN-λ 2 (**H**) in CM isolated from poly I:C-treated TOs (blue) or DOs (orange) or in mock-treated controls (gray). In (**F–H**), each symbol represents individual CM preparations. In (**F–H**), data are shown as mean ± standard deviation and significance was determined using a Mann-Whitney U-test. **p<0.01, *p<0.05; ns, not significant.

The online version of this article includes the following figure supplement(s) for figure 5:

**Figure supplement 1.** Factors induced by poly I:C treatment of TOs and DOs.

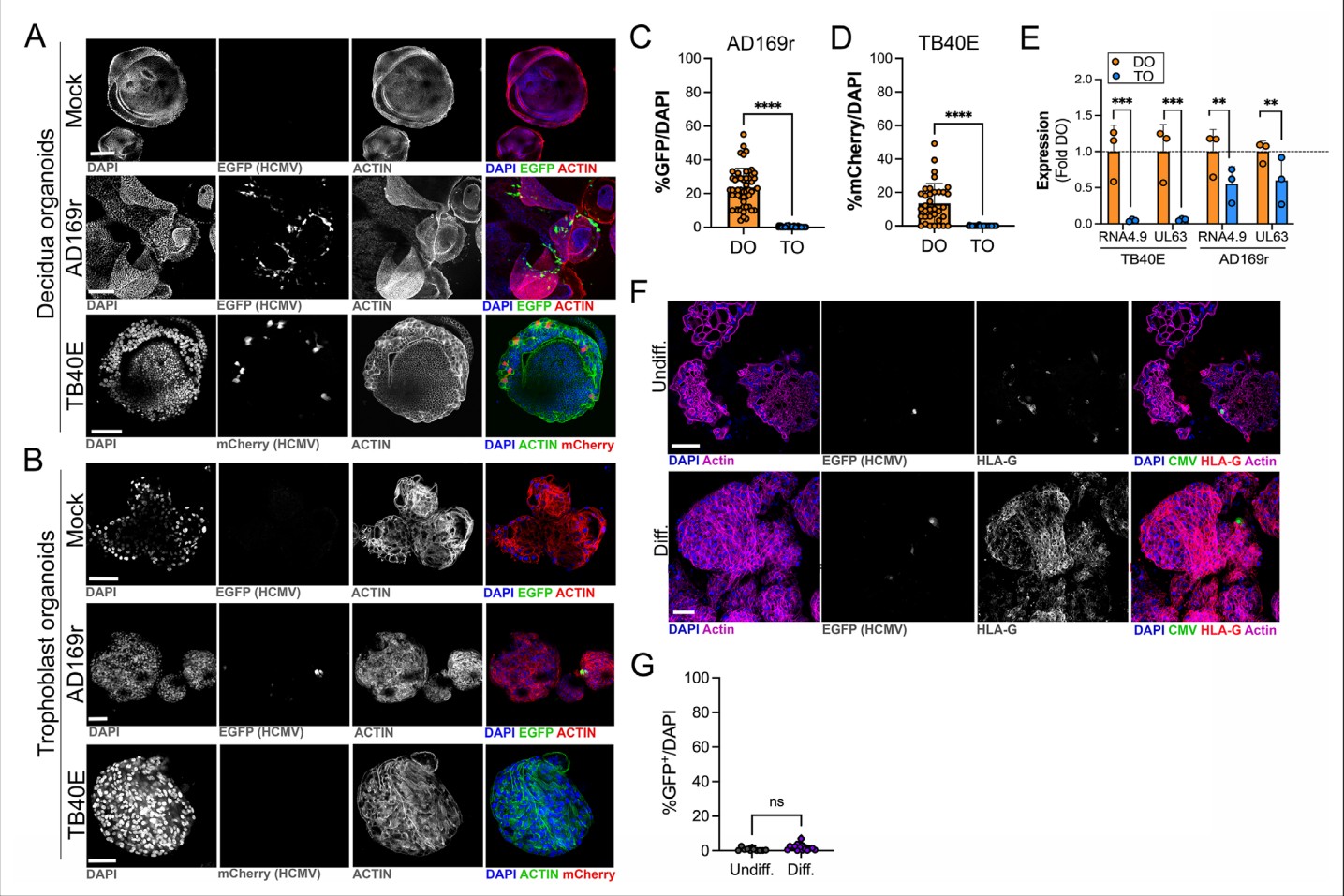

**Figure 6.** Differential susceptibility of TOs and DOs to HCMV infection. (**A, B**) Representative confocal micrographs of DOs (**A**) or TOs (**B**) infected with human CMV (HCMV) strains EGFP-tagged AD169r and mCherry-tagged TB40E for ~48 hr. Mock infected shown in top row of (**A**) and (**B**). Infected cells are positive for EGFP (AD169r, middle row) or mCherry (TB40E, bottom row). Organoids were stained for actin (red in mock- and AD169r-infected or green in TB40E infected). DAPI-stained nuclei are shown in blue. (**C, D**) Quantification of HCMV infection in DOs or TOs infected with AD169r (**C**) or TB40E (**D**). Shown is the percent HCMV positive cells as assessed by EGFP (AD169r, (**C**)) or mCherry (TB40E, (**D**)) over total number of nuclei from confocal micrographs. A minimum of 30 independent organoids taken from biological replicate experiments were quantified by automated image analysis. (**E**) Comparison of RPKM values for HCMV RNA4.9 and UL63 in TOs (blue) or DOs (orange) as assessed by bulk RNASeq. Data are shown as mean ± standard deviation as a fold of DO values. Significance was determined using a t-test. ***p<0.001, **p<0.01. Each symbol represents an independent sample. (**F**) Representative confocal micrographs of undifferentiated (top) of EVT differentiated (bottom) TOs infected with EGFP-tagged AD169r and immunostained for HLA-G (in red). DAPI-stained nuclei are shown in blue and actin is in purple. Middle panels are black and whote imaged of HCMV infection and HLA-G immunostaining. (**G**) Quantification of HCMV (AD169r strain) infection in TOs cultured under standard conditions (Undiff., left) or under conditions that promote EVT differentiation (Diff., right). Shown are the percentage of total cells that were positive for HCMV, as assessed by the presence of GFP, over total nuclei as assessed by DAPI staining. Each symbol represents an independent field from replicates, with at least 2000 total nuclei quantified under both conditions. Significance was determined using a t-test, ns, not significant. In (**A, B**), F scale bar, 50 μm. DO, decidua organoid; EVT, extravillous trophoblast; HCMV, human cytomegalovirus; TO, trophoblast organoid.

## Differential transcriptional responses of TOs and DOs to HCMV infection

Because we observed significant differences between the susceptibility of TOs and DOs to HCMV infection with strains AD169r and TB40E, we next profiled the transcriptional changes induced by HCMV infection in these organoids. Matched TOs and DOs were infected with AD169r or TB40E and the transcriptional impact of infection was assessed by bulk RNASeq transcriptional profiling. TOs responded to HCMV infection through the differential expression of 282 transcripts (AD169r strain) or 173 transcripts (TB40E strain), of which ~38% were shared between both strains of HCMV (*Figure 7A and C*). By contrast, DOs responded to HCMV infection less robustly, with 95 (AD169r

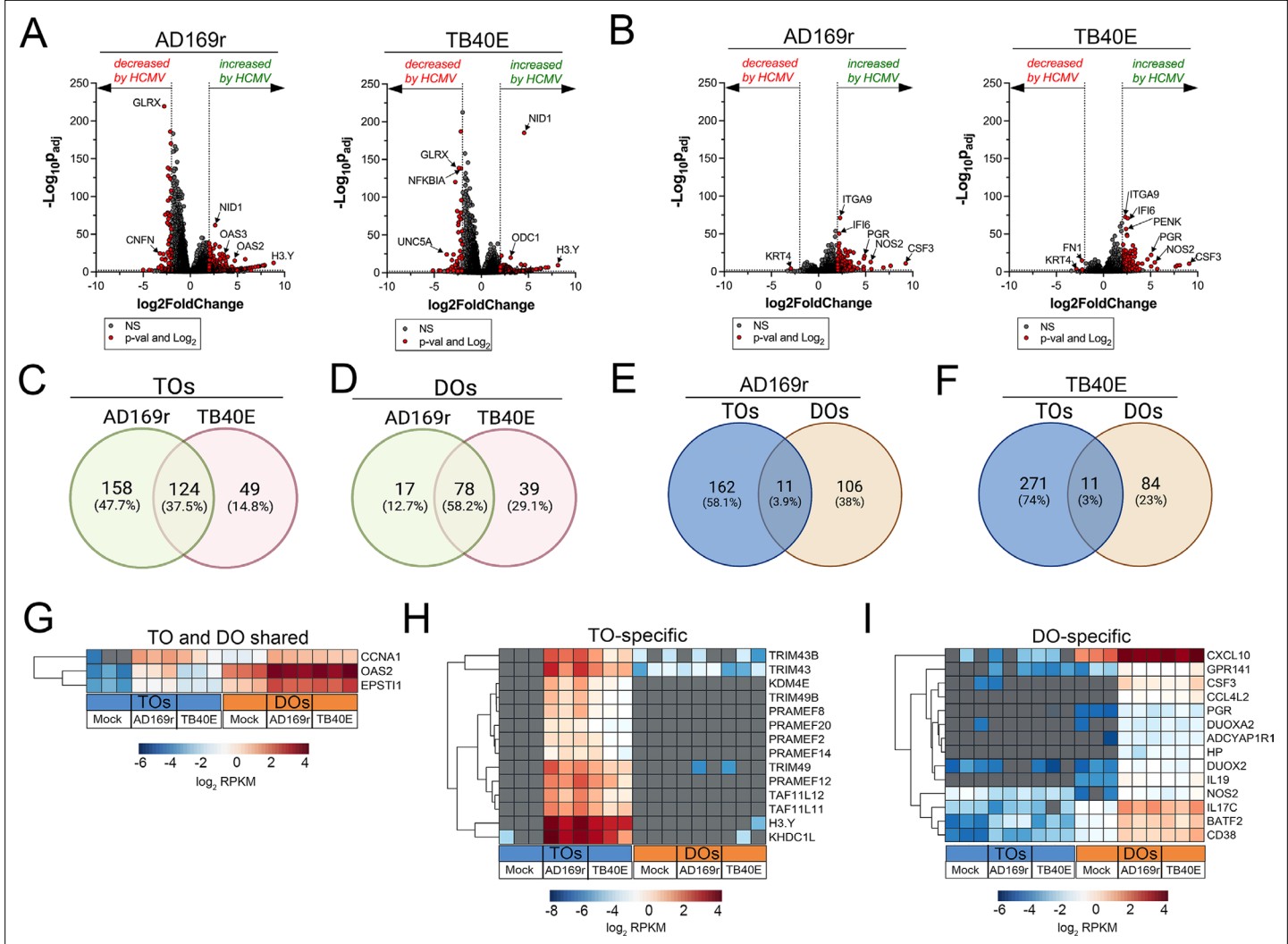

**Figure 7.** Transcriptional profiling of HCMV infected trophoblast organoids (TOs) and decidua organoids (DOs). (**A, B**) Volcano plots demonstrating differentially expressed transcripts in TOs (**A**) or DOs (**B**) infected with HCMV strains AD169r or TB40E (as denoted at top) by DeSeq2 analysis. Gray circles represent genes whose expression was not significantly changed and red denotes transcripts significantly changed by HCMV infections. Significance was set at p<0.01 and $\log_2$ fold-change of ±2. (**C, D**) Venn diagrams of differentially expressed transcripts in TOs (**C**) or DOs (**D**) infected with HCMV AD169r (green circle) or TB40E (red circle) strains. (**E, F**) Venn diagrams of the extent of overlap in differentially expressed transcripts in TOs (blue circle) or DOs (orange circles) infected with HCMV AD169r (**E**) or TB40E (**F**) strains. (**G**) Heatmap (based on $\log_2$ RPKM values) of transcripts induced by HCMV infection of both TOs (blue box, bottom) or DOs (orange box, bottom). (**H, I**) Heatmap (based on $\log_2$ RPKM values) of transcripts induced by HCMV infection specifically in TOs (**H**) or DOs (**I**). In (**G–I**), key at bottom and red indicates high levels of expression, blue indicates low levels of expression, and gray indicates no reads detected. Hierarchical clustering is on the left. HCMV, human cytomegalovirus.

strain) or 117 (TB40E strain) transcripts differentially expressed by this infection (*Figure 7B and D*). Of these differentially expressed transcripts, only 3–4% were shared between TOs and DOs infected with AD169r (*Figure 7E*) or TB40E (*Figure 7F*). Of these shared transcripts, only three were induced by both strains of HCMV in both TOs and DOs and included CCNA1 (Cyclin A1), the ISG OAS2, and EPSTI1 (Epithelial Stromal Interaction 1) (*Figure 7G*). Transcripts specifically induced by HCMV infection of TOs included H3Y1 (H3.Y Histone 1), KHDC1L (KH Domain Containing 1 Like), several members of the PRAME Family (PRAMEF2, 8, 12, 14, and 20), and TRIM49A and B (Tripartite Motif Containing 43) (*Figure 7H*). In contrast, DO-specific transcripts included several components of the nitric oxide signaling pathway including NOS2 (Nitric Oxide Synthase 2), DUOX2 (Dual Oxidase 2), and DUOXA2 (Dual Oxidase Maturation Factor 2) as well as CXCL10 and CSF3/G-CSF (*Figure 7I*).

## Differential immune secretome of TOs and DOs infected with HCMV

To define the innate immune responses to HCMV infection in TOs and DOs, we used Luminex multi-analyte profiling of >100 cytokines and chemokines in TOs and DOs infected with HCMV strains AD169r or TB40E. We found that TOs secreted relatively few cytokines in response to HCMV infection, including pentraxin-3 (PTX3), first described as an endothelial factor induced by IL-1β treatment (*Breviario et al., 1992*), IL-8, and IL-11 (*Figure 8A, C and D*, *Figure 8—figure supplement 2A*). In contrast, DOs responded to HCMV infection by secreting both proinflammatory cytokines and chemokines (e.g., CXCL10, MIP-1α, MIP-1β, and IL-6) and through the release of the type III IFN IFN-$\lambda$2 (*Figure 8B, C and E*, *Figure 8—figure supplement 2B*). In contrast to IFN-$\lambda$2, DOs did not induce any other detectable IFNs such as IFN-α2 or IFN-β or IFN-$\lambda$1 (*Figure 8—figure supplement 2C-E*).

Because we observed high levels of basal IFN-$\lambda$2 secretion in TOs, which were resistant to HCMV infection, and high induction of IFN-$\lambda$2 in DOs infected with HCMV, we next determined whether inhibiting this IFN signaling would alter the sensitivity of DOs or TOs to HCMV infection. To test this, we used the JAK-specific inhibitor ruxolitinib, which inhibits signaling downstream of type III IFN receptor activation. DOs and TOs were treated with ruxolitinib for 1 hr prior to infection with the AD169r strain of HCMV for 48 hr in the presence of ruxolitinib. We found that treatment of DOs with ruxolitinib significantly increased levels of HCMV replication, which correlated with higher levels of GFP fluorescence (*Figure 8F and G*). In contrast, treatment of TOs with ruxolitinib did not increase levels of HCMV replication over DMSO treated controls, which remained low under both ruxolitinib-treated and DMSO-treated conditions (*Figure 8G*).

To model communication between the maternal-fetal interface and to define this impact of this communication on HCMV infection, we established a co-culture model of TOs and DOs developed from matched placental tissue. To do this, equivalent numbers of TOs and DOs were pre-mixed prior to culturing and co-cultured for ~5 days prior to infection with HCMV. We confirmed that co-cultured organoids contained both TOs and DOs and distinguished between these organoids using SIGLEC-6 immunostaining, with only TOs positive for this marker (*Figure 8H*). Following the establishment of co-cultures, we infected DOs or TOs cultured alone or co-cultures and TOs and DOs with HCMV and determined the impact of this culturing on infection. We found that whereas singly cultured DOs were permissive to HCMV infection, co-culturing of DOs and TOs led to a significant reduction in HCMV infection in DOs (*Figure 8I and J*).

Taken together, these findings highlight the distinct immunoregulatory and transcriptional cascades induced by HCMV infection of maternal-derived decidual versus fetal-derived placental cells and suggest that IFNs restrict HCMV infection in decidual epithelial cells. In addition, our co-culture studies support a model whereby TO-derived factors confer protection of epithelial cells in the decidua from HCMV infection.

## Discussion

Our work establishes fetal- and maternal-derived organoids as models for defining innate immune signaling at the maternal-fetal interface. Similar to PHTs (*Bayer et al., 2016*) and mid-gestation chorionic villi (*Corry et al., 2017*), we show that TOs constitutively secrete type III antiviral IFN-$\lambda$, that act in paracrine to restrict viral infections. In contrast, although DOs did not basally secrete IFNs, they are immunologically more active under resting conditions and basally secrete cytokines and chemokines not detectable in TOs. In addition, we show that TOs and DOs respond to TLR3 activation via the induction of distinct immune regulatory networks, which includes the specific release of select chemokines such as CCL22 in TOs. Finally, we used matched TOs and DOs to explore differences in susceptibility to HCMV and to define the differential responses of the maternal decidua and fetal-derived organoids to HCMV infection. Taken together, our work suggests that cells that comprise the maternal-fetal interface defend from viral infections in a cell-type-specific manner and further highlight the unique contributions of fetal-derived trophoblasts in antiviral immunity.

The previous establishment of trophoblast and decidua-derived organoids has significantly expanded the in vitro systems that can model the complexity of the maternal-fetal interface (*Turco et al., 2017a*; *Turco et al., 2018*). These previous studies utilized tissue isolated from the first trimester of pregnancy (6–9 weeks of gestation for TOs and 8–12 weeks of gestation for DOs) (*Turco et al.,*

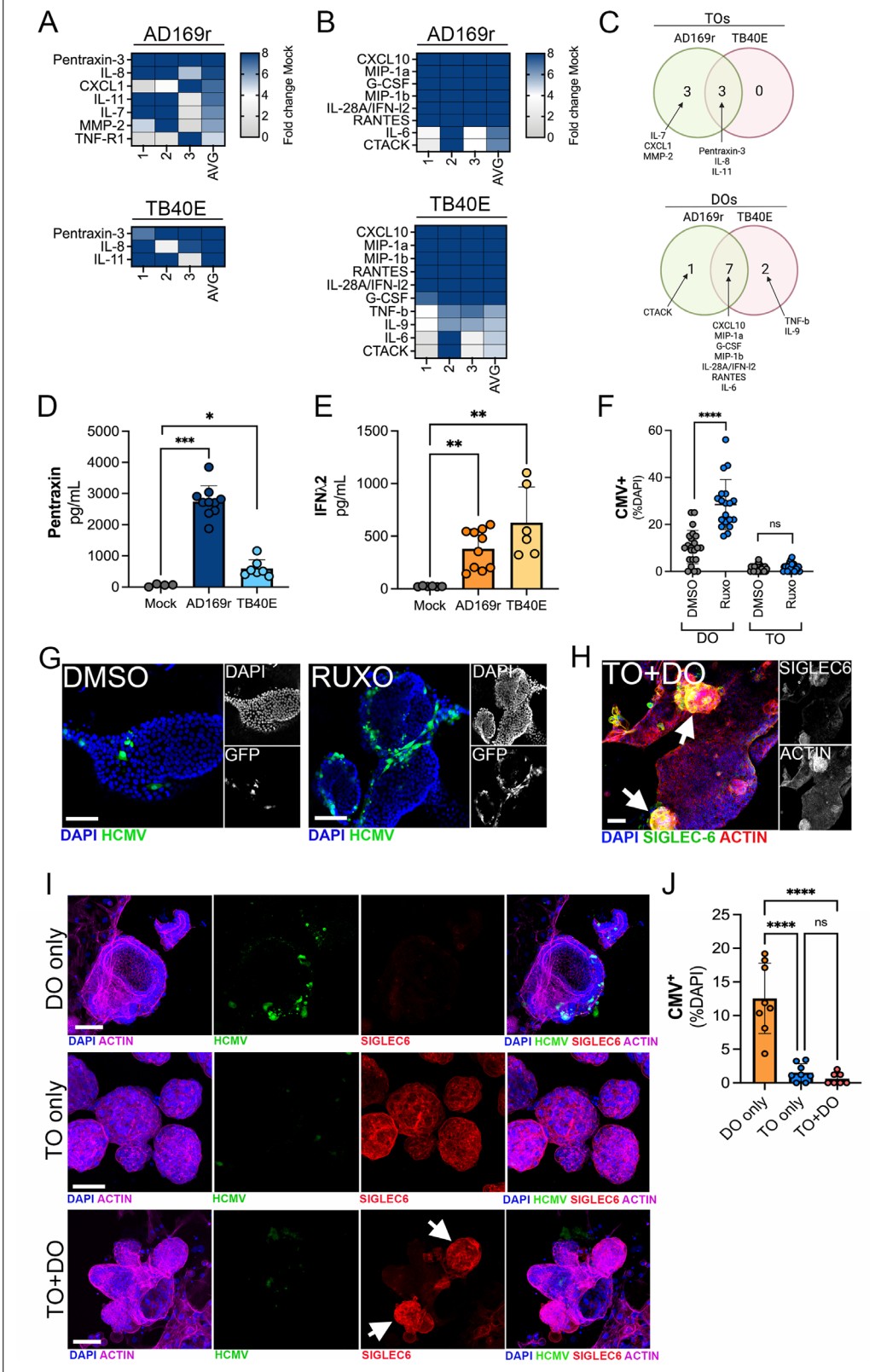

**Figure 8.** Differential innate immune responses to HCMV infection of TOs and DOs. (**A, B**) Heatmaps demonstrating the induction of factors at left (shown as fold change from mock-treated controls) from TOs (**A**) and DOs (**B**) infected with HCMV (AD169r, top or TB40E, bottom strains). Shown are factors induced >5-fold (full heatmap is shown in *Figure 8—figure supplement 2*). AVG denotes the average change in concentration of

*Figure 8 continued on next page*

*Figure 8 continued*

factors over conditioned medium (CM) isolated from three individual preparations. Dark blue denotes significantly induced factors compared with untreated controls. Gray or white denotes little to no change (scale at top right). (**C**) Venn diagrams comparing the factors induced by infection of TOs (top) or DOs (bottom) with AD169r (green) or TB40E (red). (**D**) Levels of pentraxin in TOs infected with AD169r (middle, dark blue) or TB40E (right, light blue) compared to mock-infected controls (left, gray). (**E**) Levels or IFN-$\lambda$ 2 in CM isolated from DOs infected with HCMV AD169r (middle, dark orange) or TB40E (right, light orange) or in in mock-treated controls (left, gray). Each symbol represents individual CM preparations and significance was determined using a Mann-Whitney U-test. \*\*\*p<0.001, \*\*p<0.01, \*p<0.05. (**F**) Quantification of the extent of HCMV infection in DOs (left) or TOs (right) treated with DMSO control (DMSO, gray) or with ruxolitinib (ruxo, blue) and infected with the AD169r strain of HCMV for 48 hr. Shown are the percent HCMV positive cells as assessed by GFP signal over total nuclei as assessed by DAPI staining. Each symbol represents individual fields/organoids and significance was determined using a Mann-Whitney U-test. \*\*\*\*p<0.0001, ns not significant. (**G**) Representative images from the quantification shown in (**G**) of DOs treated with DMSO control (left) or ruxolitinib (right) and infected with the GFP-tagged AD169r strain of HCMV for 48 hr. DAPI-stained nuclei are shown in blue. (**H**) Confocal micrographs of co-cultures of TOs and DOs immunostained for the trophoblast-specific marker SIGLEC-6 (in green). Actin is shown in red and a DAPI-stained nuclei are in blue. At right, black and white images of SIGLEC-6 (top) and actin (bottom). (**I**) Confocal micrographs of co-cultures singly cultured DOs (top row), TOs (middle row), or co-cultured TOs and DOs (bottom row) infected with AD169r HCMV (in green) and immunostained for the trophoblast-specific marker SIGLEC-6 (in red). Actin is shown in purple and a DAPI-stained nuclei are in blue. Black and white images are located in *Figure 8—figure supplement 1A*. (**J**) Quantification of the extent of HCMV infection in singly cultured DOs (orange, left) or TOs (blue, middle) or co-cultured TOs and DOs (pink, right) infected with the AD169r strain of HCMV for 48 hr. Shown are the percent HCMV positive cells as assessed by GFP signal over total nuclei as assessed by DAPI staining. Each symbol represents individual fields/organoids and significance was determined using a Mann-Whitney U-test. \*\*\*\*p<0.0001, ns not significant. In (**G–I**), scale bar, 50 μm. DO, decidua organoid; HCMV, human cytomegalovirus; TO, trophoblast organoid.

The online version of this article includes the following figure supplement(s) for figure 8:

**Figure supplement 1.** Co-cultures of TOs and DOs.

**Figure supplement 2.** Differential innate responses in TOs and DOs infeced with HCMV.

*2017a*; *Turco et al., 2018*), although another study successfully established DOs from full-term tissue (*Marinić et al., 2020*). Two-dimensional cultures of stem cell-derived primary trophoblasts were also obtained from the first trimester (*Okae et al., 2018*). The placenta is a dynamic organ that changes throughout gestation. Here, we show that TOs can also be generated from later stages of gestation, including from progenitor cells isolated in the second and third trimesters of pregnancy and full-term placental tissue, which required optimization from conditions required for first trimester tissue. Although the abundance of progenitor cells, as defined by the presence of Ki67 positive nuclei, declines throughout gestation, we show that full-term chorionic villi contain Ki67 positive cells at low levels and that these cells can be isolated to generate self-propagating organoids. Given that obtaining first trimester tissue can be complicated by access and/or regulatory restrictions, our study provides an additional source for the development of placental organoids from tissue obtained later in gestation. In addition, given that many congenital infections occur in the second and third trimesters of pregnancy, organoids isolated from later stages of pregnancy may better reflect clinically relevant models of congenital infections. For instance, the risk of transplacental transmission of HCMV infection from mother to fetus is only 20% in the first trimester but can reach up to 75% in the third trimester, highlighting the relevance of our organoid model for studying placental HCMV infection across gestation (*Pass and Anderson, 2014*).

Unlike many other barrier cell types such as the epithelium, human trophoblasts have the unique capacity to robustly secrete cytokines and other immunological factors without stimulation. PHTs from full-term placentas and mid-gestation chorionic villi secrete cytokines that mediate innate immune defenses against bacteria, parasites, and viruses (*Ander et al., 2018*; *Bayer et al., 2016*; *Megli et al., 2021*). Consistent with this concept, serum isolated from pregnant women often contains elevated levels of these cytokines, indicating that at least some have reached the systemic circulation (*Benyo et al., 2001*; *Liu et al., 2019*; *Megli et al., 2021*; *Vassiliadis et al., 1998*). Similarly, the decidua releases specific cytokines and alterations in these profiles may be associated with pregnancy complications including miscarriage (*Alam et al., 2008*; *Denison et al., 1998*; *Laham et al., 1997*; *Lash*

*and Ernerudh, 2015*; *Pitman et al., 2013*; *von Rango et al., 2003*). The decidua also responds to microbial infections via the release of additional cytokines that function in innate defense (*Barber et al., 2005*; *Guzeloglu-Kayisli et al., 2020*; *Qiu et al., 2009*). However, in many cases, these studies were performed on tissue explants that were multicellular in nature, thus preventing an assessment of the contributions of cytokine release from distinct cell types. Similarly, previous work utilizing tissue explants that were treated with recombinant cytokines, synthetic ligands of various innate immune pathways, or infected with pathogens are also unable to define the contributions of distinct cell types in these responses. The use of organoids allows for analyses of cellular responses to infection or innate immune activation from trophoblasts and decidual glands in primary cell models that contain isolated cell types enriched at the maternal-fetal interface. An additional strength of the TO model is that trophoblast stem cells can differentiate into all three distinct trophoblast cell types comprising the human placenta, which is not possible in many other existing models including isolation of CTBs from full-term placentas, which do not differentiate to contain EVTs.

The constitutive release of IFN-$\lambda$s from human trophoblasts is a unique phenomenon that runs counter to much of what is known about antiviral IFN signaling, which is centered on a requirement for viral sensing as a prerequisite to induce this pathway. Similar to our previous studies in PHT cells and mid-gestation chorionic villous explants (*Bayer et al., 2016*; *Corry et al., 2017*), our data show that progenitor cell-derived TOs also constitutively release IFN-$\lambda$s and protect the epithelium of the decidua from viral infections. Our studies also show that type III IFNs are further induced by activation of antiviral innate immune signaling in both TOs and DOs, with IFN-$\lambda$s being amongst the most induced cytokines detected in both organoid types. Although IFNs are released from TOs constitutively and upon induction of antiviral signaling, our studies also point to select chemokines as key components in the responses of the human placenta to viral infections. We show that the regulatory T cell (Treg) chemoattractant CCL22 is selectively released from TOs in response to TLR3 activation. We have shown previously that human PHTs and mid-gestation chorionic villi also respond to *T. gondii* infection through the selective release of CCL22 (*Ander et al., 2018*; *Rudzki et al., 2021*), suggesting that this chemokine may be key mediator of the response of the human placenta to microbial infections. Given that Treg infiltration into the decidua has been associated with adverse pregnancy outcomes including miscarriage (*Freier et al., 2015*), CCL22 induction may directly impact maternal-fetal tolerance in the context of congenital infection.

Significant work over many decades using first trimester placental explants and isolated human trophoblasts have revealed important insights into placental HCMV transmission and antiviral immunity (reviewed in *Njue et al., 2020*). Yet, a full understanding of the distinct roles of decidual versus trophoblast cells in HCMV infection has been hindered by a lack of tractable model systems of the human maternal-fetal interface. In this study, we leveraged HCMV as a model microorganism to demonstrate the utility of organoid models for studying infections at the maternal-fetal interface. Using TO and DO, we found that DOs had heightened susceptibility to HCMV infection compared to matched TOs, which is consistent with prior work in first trimester decidua explants showing that HCMV productively infects the maternal decidua (*Weisblum et al., 2015*; *Weisblum et al., 2015*). Our transcriptional and immune secretome profiling further highlights that TOs and DOs differentially respond to HCMV infection and recapitulates prior findings using isolated human placental cells and explants. CXCL10 was highly upregulated in HCMV-infected DOs, consistent with prior findings of increased CXCL10 secretion in decidua explants infected with HCMV ex vivo (*Weisblum et al., 2017*; *Weisblum et al., 2015*). IL-8 was also highly induced in HCMV-infected TOs, mirroring the increased IL-8 secretion that has been observed in response to ex vivo HCMV infection of isolated trophoblasts (*Kovács et al., 2007*) and placental explants (*Weisblum et al., 2017*). That our in vitro TO and DO infections recapitulate innate immune responses previously observed in vivo and ex vivo validates our model of matched decidua and trophoblast organoids for studying HCMV and other congenital pathogens. Moreover, our data demonstrate that type III IFNs control HCMV infection in the maternal decidua but suggest that additional factors or pathways beyond IFN signaling may restrict HCMV infection of trophoblasts. The constitutive release of type III IFNs from fetal trophoblasts has been proposed to play a direct role in the protection of maternal-derived cells from viral infections. Indeed, ablation of type III IFN signaling in the murine placenta enhances infection of ZIKV in maternal decidua tissue (*Jagger et al., 2017*). The co-culture model presented here also supports a model in which trophoblast-derived effectors, including IFN-$\lambda$, work in paracrine to suppress HCMV infection in cells

comprising the maternal decidua. Altogether, these data highlight the utility of organoid models for understanding HCMV transmission and immune defense at the maternal-fetal interface and underscore the need for more research using organoid models to disentangle the roles of decidual versus placental cells in congenital viral infections.

Our work presented here develops TOs and DOs from mid-to-late gestation placental tissue and demonstrates that these organoids can be used to study mechanisms of microbial vertical transmission and antiviral innate immune signaling. We show that TOs and DOs constitutively secrete unique cytokines and chemokines and respond to viral infections through the release of organoid-specific immunomodulatory factors. Our work also highlights the prominent role of type III IFNs in antiviral immunity at the maternal-fetal interface and demonstrates that TOs constitutively release IFN-$\lambda$s and that DOs can be induced to release IFN-$\lambda$s following viral infection. Collectively, these studies define the differential responses of decidual and placental cells comprising the maternal-fetal interface to viral infections and suggest that organoid-based models can be used to define innate immune signaling at this interface.

## Materials and methods

### Human samples

Human tissue used in this study was obtained through the UPMC Magee-Womens Hospital Obstetric Maternal & Infant Database and Biobank or from Duke University after approval was received from the University of Pittsburgh or Duke University Institutional Review Board (IRB) and in accordance with the guidelines of the University of Pittsburgh and Duke University human tissue procurement. Placental tissue used in this study was collected from the second (23rd–27th weeks) and third (35th–41st weeks) trimesters. Placental tissue was excluded for diagnosis of PPROM or maternal infection.

### Derivation and culture of TOs from human placental samples

Villous trophoblast stem/progenitor cells were isolated from fresh placental tissues as described previously with further optimization based on the characteristics of late gestation placenta villi (*Turco et al., 2018*). Briefly, villi collected from fresh placental tissue were intensively washed, then were sequentially digested with 0.2% trypsin-250 (Alfa Aesar, J63993-09)/0.02% EDTA (Sigma-Aldrich, E9884-100G) and 1.0 mg/mL collagenase V (Sigma-Aldrich, C9263-100MG) and further mechanically disrupted by pipetting up and down vigorously ~10 times with 10 mL serological pipette. Pooled digests were washed with Advanced DMEM/F12 medium (Life Technologies 12634-010) and pelleted by centrifugation, then resuspended in approximate 10 times volume of ice-cold growth-factor-reduced Matrigel (Corning 356231). Matrigel 'domes' (40 µL/well) were plated into 24-well tissue culture plates (Corning 3526), placed in a 37°C incubator to pre-polymerize for ~3 min, turned upside down to ensure equal distribution of the isolated cells in domes for another 10 min, then carefully overlaid with 500 µL prewarmed trophoblast organoid medium (TOM) consisting of Advanced DMEM/F12 (Life Technologies, 12634-010) supplemented with 1× B27 (Life Technologies, 17504-044), 1× N2 (Life Technologies, 17502-048), 10% fetal bovine serum (FBS) (vol/vol, Cytiva HyClone, SH30070.03), 2 mM GlutaMAX supplement (Life Technologies, 35050-061), 100 µg/mL Primocin (InvivoGen, ant-pm-1), 1.25 mM N-Acetyl-L-cysteine (Sigma-Aldrich, A9165), 500 nM A83-01 (Tocris, 2939), 1.5 µM CHIR99021 (Tocris, 4423), 50 ng/mL recombinant human EGF (Gibco, PHG0314), 80 ng/mL recombinant human R-spondin 1 (R&D systems, 4645-RS-100), 100 ng/mL recombinant human FGF2 (Peprotech, 100-18C), 50 ng/mL recombinant human HGF (Peprotech, 100-39), 10 mM nicotinamide (Sigma-Aldrich, N0636-100G), 5 µM Y-27632 (Sigma-Aldrich, Y0503-1MG), and 2.5 µM prostaglandin E2 (PGE2, R&D systems, 22-961-0). Cultures were maintained in a 37°C humidified incubator with 5% $CO_2$. Medium was renewed every 2–3 days. Small trophoblast organoid spheroids became visible by approximately day 12 post-isolation. Derived TOs were passaged every 5–7 days depending on their size and density. To passage, TOs were released from domes with ice-cold cell recovery solution (Corning, 354253) and then were digested in prewarmed TrypLE Express (Life Technologies, 12605-028) or Stem Pro Accutase (Life Technologies, A11105-01) at 37°C for 5–7 min. Further mechanical dissociation was achieved with an Eppendorf Explorer Plus electronic pipettes at its 'mix' function setting by pipetting up and down around 400 times through small-bore pipette tips. Dissociated TOs fragment were collected and washed by centrifuge, then resuspended in fresh ice-cold Matrigel

and replaced as domes at the desired density for continuous culture. For freezing TOs, overlaid TOM was aspirated, TOs with Matrigel were resuspended in CryoStor CS10 stem cell freezing medium (STEMCell Technologies, 07930) frozen at –80 °C and then transferred to liquid nitrogen for long-term storage. For thawing cryopreserved TOs, organoids were thawed as quickly as possible, diluted with five times volume of basic TOM containing Advanced DMEM/F12, 2 mM GlutaMAX supplement, 10 mM HEPES (Gibco, 15630-106), 1× Penicillin/Streptomycin (Lonza, 17-602E) and centrifuged to pellet. Afterward, TOs were resuspended in new ice-cold Matrigel and replated for recovery culture and passaged as described below. Trophoblast organoids used in this study were derived from 15 independent placentas (7 male and 8 female). Detailed protocols for the isolation and passaging of trophoblast organoids derived from full-term placental tissue have been posted and are available for download (*Yang and Coyne, 2022a*; *Yang and Coyne, 2022b*).

## Derivation and culture of DOs cultures from human decidua samples

Decidual gland-enriched cell suspension was acquired from fresh decidual tissue as described previously (*Turco et al., 2017b*; *Turco et al., 2017a*). Briefly, isolated decidual tissues were cut into small pieces then digested in 1.25 U/mL Dispase II (Sigma-Aldrich, D4693)/0.4 mg/mL collagenase V (Sigma-Aldrich, C-9263) solution. Glandular cells were filtered out from the digestion solution supernatant and resuspended in ice-cold Matrigel (Corning, 356231), plated as domes in 24-well plates (Corning, 3526), and finally overlaid with 500 μL prewarmed organoid Expansion Medium (ExM) with the same composition as previously described (*Turco et al., 2017a*). The growth medium was renewed every 2–3 days. Mature organoids were passaged by mechanically disruption following TrypLE Express digestion every 3–5 days. Other maintenance were similar to those as previously described (*Turco et al., 2017b*).

## Differentiation of TOs into EVT-enriched TOs

To generate EVT-enriched TOs, we adopted EVT differentiation protocols developed previously (*Okae et al., 2018*; *Sheridan et al., 2021*). TOs were passaged as described above and plated directly into eight-well chamber slides (Millicell, EZslide, Millipore) pre-coated with ~15 μL Matrigel. After plating, organoids were maintained in TOM as described above for 3–4 days, then switched to EVT differentiation media 1 (EVT m1: advanced DMEM/F12 supplemented with 2 mM L-glutamine, 0.1 mM 2-mercaptoethenol, 0.5% [vol/vol] penicillin/streptomycin, 0.3% [vol/vol] BSA, 1% [vol/vol] ITS-X supplement, 100 ng/mL NRG1 [Cell Signaling Technology, 5218SC]), 7.5 μM A83-01, 2.5 μM Y27632, and 4% (vol/vol) Knockout Serum Replacement (Gibco, 10828010), for 9 days. EVT m1 was renewed every 2 days for the initial 4 days then daily for the remaining culture period. Subsequently, organoids were switched to EVT m2 with the same recipe as EVT m1, but lacking NRG1 for a further 3–4 days, with daily media changes. HCMV infection of EVT-enriched TOs in eight-well chamber slide was performed as described as described below.

## Co-cultures of matched TOs and DOs

Matched lines of TOs and DOs were passaged regularly as described above. After passaging, equivalent numbers of TOs and DOs were combined, centrifuged, then resuspended with TOM supplemented with 5 μM Y27632. Mixed TOs and DOs were plated directly into eight-well chamber slides pre-coated with ~15 μL Matrigel and cultured for ~5 days prior to HCMV infection performed as described below. The presence of TOs was confirmed using OTC rapid hCG test strips, with only DOs co-cultured with TOs testing positive (*Figure 8—figure supplement 1B*).

## Generation of TO and DO conditioned media

Overlaid TOM and ExM were collected every 2–4 days based on the growth status of TOs and DOs. Media from at least 10 time points for each established organoid lines were collected as TO-CM or DO-CM. For TO CM, media were collected starting at ~4 days after passaging, or when the majority of organoids were 100–200 μm in size. The levels of hCG in collected CM were measured by Luminex assays or rapid hCG test strips prior to use. CM with low levels of hCG were not included in subsequent experiments. For DO CM, media were collected ~2 days after passaging, or when most organoids had formed obvious cystic structure of at least 200 μm. In most experiments, the use of TO and DO CM from matched tissue was used, although there did not appear to be any significant differences

in CM from unmatched donor tissue. Non-conditioned medium (NCM) was TOM or ExM (described above) that had not been exposed to TOs or DOs, respectively. All experiments were performed with CM from at least three independent TO and DO lines.

## Cells and viral infections

Human osteosarcoma U2OS (ATCC HTB-96) and Vero (ATCC CCL-81) cells were grown in Dulbecco's minimal essential medium (DMEM) with 10% FBS and 1% antibiotic. HEK-Blue IFN-$\lambda$ reporter cells (hkb-ifnl) were purchased from InvivoGen and cultured according to the manufacturer's instructions. Cell lines were tested for mycoplasma at least every 2–4 weeks using the MycoStrip Mycoplasma Detection Kit (Invivogen, rep-mys-50). ZIKV (Paraiba/2015, provided by David Watkins, University of Miami) was propagated in Vero cells. Viral titers were determined by fluorescent focus assay as previously described (*Payne et al., 2006*) using recombinant anti-double-stranded RNA monoclonal antibody J2 (provided by Abraham Brass, University of Massachusetts) in Vero cells. Infection was determined by qRT-PCR as described below. To assess antiviral activity, U2OS cells were exposed to TO or DO CM, or matched NCM, 24 hr prior to infection with ZIKV Paraiba/2015 for 72 hr. GFP-tagged AD169r (BADrUL131-Y4, a gift from T. Shenk; *Wang and Shenk, 2005*) and mCherry-tagged TB40/E (a gift from N. Moorman; *O'Connor and Shenk, 2011*) were propagated by infecting HFF cells (ATCC SCRC-1041) (MOI=0.01) followed by incubation for ~14 days until 90–95% of cells showed cytopathic effect. Cells and supernatant were harvested, filtered (0.45 μm), and then concentrated on a 20% sucrose cushion (20,000 rpm) before the virus pellet was resuspended in sterile media before titration on HFFs and retinal epithelial ARPE-19 cells (ATCC CRL-2302). HCMV infection of TOs and DOs was performed using 3–4×10^5 PFU virus per well of an 8-well chamber slide or 1x10^6 per well of a 24-well plate for 48 hr. For infections, TOs and DOs were plated directly onto 15 μL of a Matrigel coating in each well of an 8-well chamber slide (Millicell, EZslide, Millipore) or onto 20 μL of Matrigel coating in 24-well plates and cultured for 5–7 days prior to infection. After 48 hr, CM was harvested and stored at –80°C until Luminex profiling and cells were fixed with paraformaldehyde (PFA) for 15 min before staining for immunofluorescence microscopy, as detailed below.

## Immunofluorescence microscopy

TOs and DOs were released from Matrigel domes with 1 mL of cell recovery solution (Corning, 354253) per well without disrupting their 3D architecture, then were fixed in 4% PFA for 30 min at room temperature, followed by 0.5% Triton X-100/phosphate-buffered saline [PBS] to permeabilize for 30 min at 4°C. Organoids were washed and blocked in 5% (v/v) goat serum/0.1% (v/v) Tween-20 in PBS for 15 min at room temperature and then incubated with primary antibodies in the above-described blocking solution at 4°C overnight. Organoids were then washed with PBS and then incubated for 1–2 hr at room temperature with Alexa Fluor-conjugated secondary antibodies (Invitrogen). Organoids were washed again with PBS and mounted in Vectashield (Vector Laboratories) containing 4',6-diamidino- 2-phenylindole (DAPI) and were transferred onto microscope slides with large-orifice 200 μL tips (Thermo Fisher Scientific, 02707134). For staining performed in eight-well chamber slides, the same protocol described above was used, but the releasing of organoids from Matrigel was omitted. The following antibodies or reagents were used: cytokeratin-19 (Abcam, ab9221 and ab52625), E-cadherin (Invitrogen, PA5-85088), EpCAM (Abcam, ab223582), Mucin 5AC (Abcam, ab3649), SIGLEC6 (Abcam, ab262851), HLA-G (Abcam, ab52454 and ab283260), Ki67 (550609, BD Biosciences), SDC-1 (Abcam, ab128936), J2 double-stranded RNA antibody as described previously (*Bayer et al., 2016*), Alexa Fluor 594-conjugated phalloidin (Invitrogen, A12381), Alexa Fluor 633-conjugated phalloidin (Invitrogen, A22284), Alexa Fluor Plus 488 Goat anti-Mouse IgG secondary antibody (Invitrogen, A32723), Alexa Fluor 594 Goat anti-Mouse IgG secondary antibody (Invitrogen, A11032), Alexa Fluor 488 Goat anti-Rabbit IgG secondary antibody (Invitrogen, A11034), and Alexa Fluor 594 Goat anti-Rabbit IgG secondary antibody (Invitrogen, A11037). Images were captured using Zeiss 880 Airyscan Fast Inverted confocal microscope and contrast-adjusted in Photoshop or Fiji (*Schindelin et al., 2012*). For whole Matrigel dome images, scans were performed using a 10× objective on an inverted IX83 Olympus microscopy with a motorized XY-stage (Prior) and tiled images automatically generated by CellSens (Olympus). Image analysis and three-dimensional image reconstruction were performed using Imaris version 9.2.1 (Oxford Instruments).

## Immunohistochemistry

Tissue sections were deparaffinized with xylene and rehydrated with decreasing concentrations of ethanol (100%, 95%, and 80%), then washed with ddH₂0. Antigen retrieval was performed with slides submerged in 10 mM citrate buffer (pH 6.0) and heated in a steamer for 90°C for 20 min. Slides were cooled to room temperature and incubated with 6% $H_2O_2$ in methanol for 30 min. Following washing in 0.1% PBS-T (PBS, 0.1% Tween 20), slides were incubated in Avidin blocking solution for 15 min, followed by subsequent blocking in Biotin blocking solution for 15 min (Vector Laboratories, SP-2001). Following washing in 0.1% PBS-T, slides were then incubated with serum-free Protein Block (Abcam, ab156024) for 10 min. Sections were incubated with primary antibodies (cytokeratin-19; Abcam, ab9221) and Ki67 (550609, BD Biosciences) diluted 1:250 in PBS-T overnight in a humidified chamber at 4°C. Next, slides were washed with PBS-T and incubated with secondary antibody (Biotinylated Goat Anti-Rabbit or Mouse IgG, Vector Biolaboratories BA-1000 and BA-9200) for 30 min, washed, and then incubated with avidin/biotin-based peroxidase (Vector Laboratories, Vectastain Elite ABC HRP Kit, PK-6100) for an additional 30 min. Following washes in PBT-T, sections were incubated with DAB substrate (Vector Laboratories, SK-4100) for ~5 min. Slides were washed with ddH₂0 and then counterstained with hematoxylin for 1 min, thoroughly rinsed with $H_2O$, and incubated in 0.1% sodium bicarbonate in $H_2O$ for 5 min. Slides were then dehydrated with increasing concentrations of ethanol, cleared with xylene, and mounted with Vectamount Permanent Mounting Medium (Vector Laboratories, H-5000). Images were captured on an IX83 inverted microscope (Olympus) using a UC90 color CCD camera (Olympus).

## RNA extraction, cDNA synthesis, and quantitative PCR

Total RNA was extracted with the Sigma GenElute total mammalian RNA miniprep kit following the manufacturer's instruction and using the supplementary Sigma DNase digestion. RNA quality and concentration were determined using a Nanodrop ND-1000 Spectrophotometer. Total RNA was reverse-transcribed with the iScript cDNA synthesis kit (Bio-Rad) following the manufacturer's instructions. Quantitative PCR was performed using the iQ SYBR Green Supermix (Bio-Rad, 1708882) on a CFX96 Touch Real-Time PCR Detection System (Bio-Rad). Gene expression was determined based on a $\Delta C_T$ method normalized to actin. The expression level of C19MC miRNAs hsa-miR-516b-5p, hsa-miR-517a-3p, hsa-miR-525-5p, and reference gene RNU6 was quantified by using the MystiCq MicroRNA Quantitation System (Millipore Sigma) which includes miRNAs Poly(A) tailing, cDNA synthesis (MIRRT), and miRNA qPCR (MIRRM00). MicroRNAs were first polyadenylated in a poly(A) polymerase reaction, then reverse transcribed using an oligo-dT adapter primer according to the manufacturer's protocol (Sigma-Aldrich, MIRRT). Individual microRNAs (microRNA-516b-5p, microRNA-517a-3p, and microRNA-525-5p) were quantified in real-time SYBR Green RT-qPCR reactions with the specific MystiCq microRNA qPCR Assay Primer (MIRAP00475, MIRAP00477, and MIRAP00512) and the MystiCq Universal PCR Primer (MIRUP). Cq readings were normalized to the RNU6 internal reference by the $2^{-\Delta Cq}$ method.

## RNAseq

For RNAseq analysis, RNA was isolated from organoids as described above. Purified Total RNA was verified by Thermo Scientific NanoDrop one. All RNA samples submitted for bulk RNA-seq were further evaluated for their RIN number (>9) prior to library preparation by the Duke Center for Genomic and Computational Biology (GCB) using the TruSeq stranded total RNA prep kit (Illumina). Sequencing was performed on a NovaSeq 6000 or NextSeq500 by using 150 bp paired-end sequencing. The reads were aligned to the human reference genome (GRCh38) using QIAGEN CLC Genomics (v20). DESeq2 (*Love et al., 2014*) was used to normalized count data and to perform differential gene expression analysis using a significance cutoff of 0.01 and a fold change cutoff of $\log_2$ ±2. Heatmaps and hierarchical clustering were performed in R using pheatmap package in R and were based on RPKM (reads per kilobase million). Mapping to HCMV genomes were performed in CLC Genomics (v20) using either AD169r (accession FJ527563) or TB40E (accession MW439039) reference sequences. Files associated with RNA-seq studies have been deposited into Sequence Read Archive under accession number SUB11885513. Previous data sets used in this study include those from PHT cells (accession PRJNA305417 and PRJNA388219) (*Bayer et al., 2016*; *Corry et al., 2017*; *Van Twisk et al., 2017*), first trimester placental tissue (PRJEB38810), and full-term tissue (PRJNA721668) (*Lu-Culligan et al.,*

*2021*). Principal component analyses were performed using pcaExplorer in R (*Marini and Binder, 2019*) or GraphPad Prism. Volcano plots were generated using the EnhancedVolcano package in R (*Blighe et al., 2021*) or in Graphpad Prism version 9.

## Luminex assays

All sample processing was performed in duplicate and each experiment was performed with at least three biological replicates. Luminex assays were performed using the following kits according to the manufacturer's instructions: Bio-Plex Pro Human Inflammation Panel 1, 37-Plex kit (171AL001M; Bio-Rad) Bio-Plex Human Chemokine Panel, 40-plex (171AK99MR2; Bio-Rad), Bio-Plex Pro Human Cytokine Screening Panel, 48-Plex kit (12007283; Bio-Rad), hCG Human ProcartaPlex Simplex Kit (Thermo Fisher Scientific, EPX010-12388-901), and CA125 (Mucin-16) Human ProcartaPlex Simplex Kit (EPX010-12437-901). Plates were read on MAGPIX (EMD Millipore) or Bio-Plex 200 (Bio-Rad) Luminex machines and analyzed using xPONENT (EMD Millipore) or Bio-Plex (Biorad) software.

## IFN-λ activity reporter assay

For IFN-λ activity detection from CM, HEK-Blue IFN-λ reporter cells were used according to the manufacture's protocol. Briefly, IFN-λ reporter cells were plated into a 96-well plate, then CM was added and incubated at 37°C for 24 hr. Supernatants (20 µL) were collected and then transferred into a 96-well plate and activity measured using QUANTI-Blue Solution (InvivoGen, rep-qbs) after incubating at 37°C for 2 hr using a spectrophotometer at 655 nm. Each CM preparation was run in triplicate.

## Poly (I:C) and ruxolitinib treatment

For Poly (I:C) treatment, TOs or DOs were incubated with 10 µg/mL Poly (I:C) (Invivogen, tlrl-pic) diluted in organoid complete growth media for ~24 hr, then supernatant was collected for Luminex-based multianalyte profiling or RNA isolated as described for qRT-PCR or RNASeq. For ruxolitinib (Invivogen, tlrl-rux) treatment, TOs or DOs plated as previously described for HCMV infections were pre-treated with 20 µM ruxolitinib in appropriate complete growth media for 1 hr at 37°C, then were infected with GFP-tagged AD169r in the presence of ruxolitinib for 48 hr.

## Statistics and reproducibility

All experiments reported in this study have been reproduced using independent samples (tissues and organoids) from multiple donors. All statistical analyses were performed using Prism software (GraphPad Software). Data are presented as mean ± SD, unless otherwise stated. Statistical significance was determined as described in the figure legends. Parametric tests were applied when data were distributed normally based on D'Agostino-Pearson analyses; otherwise, nonparametric tests were applied. P values <0.05 were considered statistically significant, with specific p values noted in the figure legends.

# Acknowledgements

The authors thank Jon Boyle (University of Pittsburgh) for careful review of the manuscript and the Duke University School of Medicine for the use of the Sequencing and Genomic Technologies Shared Resource, which provided RNASeq services. This project was supported by NIHAI145828 (CBC).

# Additional information

## Funding

| Funder | Grant reference number | Author |
|---|---|---|
| National Institute of Allergy and Infectious Diseases | NIHAI145828 | Carolyn B Coyne |

The funders had no role in study design, data collection and interpretation, or the decision to submit the work for publication.

## Author contributions
Liheng Yang, Eleanor C Semmes, Conceptualization, Data curation, Formal analysis, Validation, Investigation, Visualization, Methodology, Writing - original draft, Writing - review and editing; Cristian Ovies, Data curation, Formal analysis, Writing - review and editing; Christina Megli, Jennifer B Gilner, Resources; Sallie Permar, Resources, Methodology; Carolyn B Coyne, Conceptualization, Data curation, Formal analysis, Supervision, Funding acquisition, Validation, Investigation, Visualization, Methodology, Writing - original draft, Project administration, Writing - review and editing

## Author ORCIDs
Liheng Yang ![ORCID] http://orcid.org/0000-0001-6842-086X
Carolyn B Coyne ![ORCID] http://orcid.org/0000-0002-1884-6309

## Decision letter and Author response
Decision letter https://doi.org/10.7554/eLife.79794.sa1
Author response https://doi.org/10.7554/eLife.79794.sa2

---

# Additional files

## Supplementary files
• MDAR checklist

## Data availability
Sequence data have been deposited into Sequence Read Archives, PRJNA869668, PRJNA869760, and PRJNA875665.

The following datasets were generated:

| Author(s) | Year | Dataset title | Dataset URL | Database and Identifier |
|---|---|---|---|---|
| Yang L, Semmes EC, Ovies C, Megli C, Permar S, Gilner JB, Coyne CB | 2022 | Homo sapiens Raw sequence reads | https://www.ncbi.nlm.nih.gov/bioproject/?term=PRJNA869668 | NCBI BioProject, PRJNA869668 |
| Yang L, Semmes EC, Ovies C, Megli C, Permar S, Gilner JB, Coyne CB | 2022 | BioProject | https://www.ncbi.nlm.nih.gov/bioproject/?term=PRJNA869760 | NCBI BioProject, PRJNA869760 |
| Yang L, Semmes EC, Ovies C, Megli C, Permar S, Gilner JB, Coyne CB | 2022 | Bulk RNASeq from decidua and trophoblast organoids from human full-term placental tissue. | https://www.ncbi.nlm.nih.gov/bioproject/?term=PRJNA875665 | NCBI BioProject, PRJNA875665 |

The following previously published datasets were used:

| Author(s) | Year | Dataset title | Dataset URL | Database and Identifier |
|---|---|---|---|---|
| Coyne CB, Corry J, Arora N, Good CA, Sadovsky Y | 2015 | Transcriptome comparisons between two-dimensional and three-dimensional cultured JEG-3 trophoblasts, primary human trophoblast cells, and human brain microvascular endothelial cells (HBMEC) | https://www.ncbi.nlm.nih.gov/bioproject/?term=PRJNA305417 | NCBI BioProject, PRJNA305417 |

*Continued on next page*

*Continued*

| Author(s) | Year | Dataset title | Dataset URL | Database and Identifier |
|---|---|---|---|---|
| Bradley AJ, Lurain NS, Ghazal P, Trivedi U, Cunningham C, Baluchova K, Gatherer D, Wilkinson GW, Dargan DJ, Davison AJ | 2013 | Human herpesvirus 5 strain AD169, complete genome | https://www.ncbi. nlm.nih.gov/nuccore/ FJ527563.1 | NCBI GenBank, FJ527563 |
| Al Qaffas A, Camiolo S, Vo M, Aguiar A, Davison AJ, Hertel L, McVoy MA | 2021 | Human betaherpesvirus 5 strain TB40/EE, complete genome | https://www.ncbi. nlm.nih.gov/nuccore/ MW439039 | NCBI GenBank, MW439039 |
| Coyne CB, Corry J, Arora N, Good CA, Sadovsky Y | 2017 | Organotypic models of type III interferon-mediated protection from Zika virus infections at the maternal-fetal interface | https://www.ncbi.nlm. nih.gov/bioproject/? term=PRJNA388219 | NCBI BioProject, PRJNA388219 |
| University of Cambridge | 2021 | RNA-seq of human first and second trimester placenta | https://www.ncbi.nlm. nih.gov/bioproject/? term=PRJEB38810 | NCBI BioProject, PRJEB38810 |
| Lu-Culligan A | 2021 | SARS-CoV-2 infection in pregnancy is associated with robust inflammatory response at the maternal-fetal interface (human) | https://www.ncbi.nlm. nih.gov/bioproject/? term=PRJNA721668 | NCBI BioProject, PRJNA721668 |

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

# Appendix 1

## Appendix 1—key resources table

| Reagent type (species) or resource | Designation | Source or reference | Identifiers | Additional information |
|---|---|---|---|---|
| Other | Matrigel | Corning | 356231 | Growth-factor-reduced for organoid growth. |
| Other | TrypLE Express | Gibco | 12605–028 | For organoid dissociation during passaging. |
| Other | Stem Pro Accutase | Gibco | A11105-01 | For organoid dissociation during passaging. |
| Other | CryoStor CS10 stem cell freezing medium | STEMCell Technologies | 07930 | For organoid cryopreservation. |
| Other | Trypsin-250 | Alfa Aesar | Alfa Aesar | For stem cell isolation from tissue. |
| Other | Collagenase V | Sigma-Aldrich | C9263-100MG | For stem cell isolation from tissue. |
| Other | recombinant human EGF | Gibco | PHG0314 | For organoid growth media. |
| Other | recombinant human R-spondin 1 | R & D systems | 4645-RS-100 | For organoid growth media. |
| Other | recombinant human FGF2 | Peprotech | 100–18 C | For organoid growth media. |
| Other | recombinant human HGF | Peprotech | 100–39 | For organoid growth media. |
| Other | ITS-X supplement | Gibco | 51500–056 | For organoid growth media. |
| Other | KO serum replacement | ThermoFisher | 10828010 | For organoid growth media. |
| Other | NRG1 | Cell Signaling | 5218SC | For organoid growth media. |
| Other | A83-01 | Tocris | 2939 | For organoid growth media. |
| Other | Y27632 | Sigma or STEMCell Technologies | Y0503 or 72304 | For organoid growth media. |
| Other | nicotinamide | Sigma | N0636 | For organoid growth media. |
| Other | FBS | Cytiva HyClone | SH30070.03 | For organoid growth media. Heat inactivated at 50 °C for 30 min. |
| Other | Advanced DMEM/F12 medium | Life Technologies | 12634–010 | For organoid growth media. |
| Other | 24-well tissue culture plates | Corning | 3526 | For organoid plating. |
| Cell line (human) | HEK-Blue IFN-$\lambda$ reporter cells | InvivoGen | hkb-ifnl | |
| Other | HCMV AD169r BADrUL131-Y4 | T. Shenk | PMID: 16051825 | |
| Other | HCMV TB40/E | N. Moorman | PMID: 21307184 | |
| Other | 8-well chamber slide | Millipore | PEZGS0816 | For organoid plating. |
| Other | Cell recovery solution | Corning | 354253 | |
| Other | Vectashield | Vector Laboratories | H-1200–10 | Mounting medium for immunofluorescence. |
| Antibody | Mouse monoclonal Cytokeratin-19 | Abcam | ab9221 | (Immunofluorescence, 1:500) |
| Antibody | Rabbit monoclonal Cytokeratin-19 | Abcam | ab52625 | (Immunofluorescence, 1:500) |

*Appendix 1 Continued on next page*

*Appendix 1 Continued*

| Reagent type (species) or resource | Designation | Source or reference | Identifiers | Additional information |
|---|---|---|---|---|
| Antibody | Rabbit polyclonal E-cadherin | Invitrogen | PA5-85088 | (Immunofluorescence, 1:500) |
| Antibody | Rabbit monoclonal EpCAM | Abcam | ab223582 | (Immunofluorescence, 1:500) |
| Antibody | Mouse monoclonal Mucin 5AC | Abcam | ab3649 | (Immunofluorescence, 1:500) |
| Antibody | Rabbit polyclonal SIGLEC6 | Abcam | ab262851 | (Immunofluorescence, 1:500) |
| Antibody | Mouse monoclonal HLA-G | Abcam | ab52454 | (Immunofluorescence, 1:500) |
| Antibody | Rabbit monoclonal HLA-G | Abcam | ab283260 | (Immunofluorescence, 1:500) |
| Antibody | Mouse monoclonal Ki67 | BD Biosciences | 550609 | (Immunofluorescence, 1:500; Immunohistochemistry, 1:200) |
| Antibody | Rabbit monoclonal SDC-1 | Abcam | ab128936 | (Immunofluorescence, 1:500) |
| Other | Alexa Fluor 594–conjugated phalloidin | Invitrogen | A12381 | For actin staining. (Immunofluorescence 1:1000) |
| Other | Alexa Fluor 633–conjugated phalloidin | Invitrogen | A22284 | For actin staining. (Immunofluorescence 1:1000) |
| Other | Alexa Fluor Plus 488 Goat anti-Mouse IgG secondary antibody | Invitrogen | A32723 | Secondary antibody for immunofluorescence. (Immunofluorescence 1:1000) |
| Other | Alexa Fluor 594 Goat anti-Mouse IgG secondary antibody | Invitrogen | A11032 | Secondary antibody for immunofluorescence. (Immunofluorescence 1:1000) |
| Other | Alexa Fluor 488 Goat anti-Rabbit IgG secondary antibody | Invitrogen | A11034 | Secondary antibody for immunofluorescence. (Immunofluorescence 1:1000) |
| Other | Alexa Fluor 594 Goat anti-Rabbit IgG secondary antibody | Invitrogen | A11037 | Secondary antibody for immunofluorescence. (Immunofluorescence 1:1000) |
| Software, algorithm | Fiji | N/A | https://imagej.net/software/fiji/ | For image analysis, including three-dimensional reconstructions. |
| Software, algorithm | CellSens | Olympus | | N/A |
| Software, algorithm | Imaris | Oxford Instruments | | Version 9.2.; for image analysis, including three-dimensional reconstructions. |
| Software, algorithm | CLC Genomics | Qiagen | | Version 20 |
| Software, algorithm | GraphPad Prism | Dotmatics | | Version 9 |
| Other | Vectastain Elite ABC HRP Kit | Vector Laboratories | PK-6100 | For immunohistochemistry. |
| Other | Biotinylated Goat Anti-Rabbit | Vector Laboratories | BA-1000 | For immunohistochemistry. (Immunohistochemistry, 1:250) |
| Other | Biotinylated Goat Anti-Mouse | Vector Laboratories | BA-9200 | For immunohistochemistry. (Immunohistochemistry, 1:250) |
| Other | DAB substrate | Vector Laboratories | SK-4100 | For immunohistochemistry. |
| Other | Vectamount Permanent Mounting Medium | Vector Laboratories | H-5000 | For immunohistochemistry. |
| Commercial assay, kit | GenElute total mammalian RNA miniprep kit | Sigma | RTN70-1KT | With on-column DNAse treatment |

*Appendix 1 Continued on next page*

*Appendix 1 Continued*

| Reagent type (species) or resource | Designation | Source or reference | Identifiers | Additional information |
|---|---|---|---|---|
| Commercial assay, kit | iScript cDNA synthesis kit | Biorad | 1708891 | N/A |
| Commercial assay, kit | iQ SYBR Green Supermix | Biorad | 1708882 | N/A |
| Software, algorithm | DeSeq2 | M.Love et al | PMID: 25516281 | https://bioconductor.org/packages/release/bioc/html/DESeq2.html |
| Software, algorithm | pheatmap | Raivo Kolde | | https://cran.r-project.org/web/packages/pheatmap/pheatmap.pdf |
| Software, algorithm | pcaExplorer | Marini and Binder | PMID: 31195976 | https://bioconductor.org/packages/release/bioc/html/pcaExplorer.html |
| Commercial assay, kit | Bio-Plex Pro Human Inflammation Panel 1 | Biorad | 171AL001M | |
| Commercial assay, kit | Bio-Plex Human Chemokine Panel, 40-plex | Biorad | 171AK99MR2 | |
| Commercial assay, kit | Bio-Plex Pro Human Cytokine Screening Panel, 48-Plex kit | Biorad | 12007283 | |
| Commercial assay, kit | hCG Human ProcartaPlex Simplex Kit | Thermo Fisher | EPX010-12388-901 | |
| Commercial assay, kit | CA125/Mucin-16 Human ProcartaPlex Simplex Kit | Thermo Fisher | EPX010-12437-901 | |
| Chemical compound, drug | Poly(I:C) HMW | Invivogen | tlrl-pic | |
| Chemical compound, drug | Ruxolitinib | Invivogen | tlrl-rux | |
| Biological sample | Primary human trophoblasts from full term placentas | PMID: 27066743 | SRP067137 | Bulk RNASeq from primary human trophoblasts. |
| Biological sample | Primary human trophoblasts from full term placentas | PMID: 28784796 | SRP109039 | Bulk RNASeq from primary human trophoblasts. |
| Biological sample | First trimester placental tissue | University of Cambridge | PRJEB38810 | Bulk RNASeq from first trimester placental tissue. |
| Biological sample | Full-term placental tissue | PMID: 33969332 | PRJNA721668 | Bulk RNASeq from ffull-term placental tissue. |

