## [Editor Report]

Yang et al., provide a scientifically sound and compelling manuscript characterizing mid-to-late gestation trophoblast and decidual organoids as ex vivo models to study vertically transmitted microbial infections, using human cytomegalovirus as a model pathogen. They demonstrate organoids have tissue-specific immunological responses and susceptibilities to viral infection.

---

## [Decision Letter]

**Decision letter after peer review:**

Thank you for submitting your article "Innate immune signaling in trophoblast and decidua organoids defines differential antiviral defenses at the maternal-fetal interface" for consideration by *eLife*. Your article has been reviewed by 2 peer reviewers, and the evaluation has been overseen by a Reviewing Editor and Sara Sawyer as the Senior Editor. The following individual involved in review of your submission has agreed to reveal their identity: Rowan Karvas (Reviewer #2).

The reviewers agree that this is a strong and impactful manuscript describing the establishment and characterization of mid-to-late gestation trophoblast and decidual organoids and their use in studying viral infection. Several points need to be addressed to support conclusions of which the essential revisions are pointed out below.

Essential revisions:

(1) There is only a single readout for viral infection with quantification being % infected organoid (which could include organoids with only a single cell or hundreds of cells infected). A more fine-tuned quantification seems necessary given the conclusions of the manuscript.

(2) More convincing experimental data that EVTs contained in TOs are not susceptible should be included. The reviewer suggests: "The authors should differentiate their organoids into EVT organoids as described in Sheridan et al., Nature Protocols 2020 and try infection with their fluorophore-tagged hCMV viruses. Spontaneously formed pre-EVT within TOM-treated organoids are not very mature compared to EVTs within properly differentiated EVT organoids. They could try infection and whole mount image, OR dissociate the organoids and run a flow cytometry experiment to quantify the proportion of infected cells within their EVT organoids and TOM (cytotrophoblastic and syncytiotrophoblastic) organoids. “

(3) A weakness in this work is that there was no characterization of the resulting organoids proving that they indeed transcriptionally or epigenetically model a second-third trimester timepoint” or really any experimental comparison with first trimester to see how gestation age is reflected/odellin.

*Reviewer #1 (Recommendations for the authors):*

– The introduction should include a description of trophoblasts organoid orientation and what the decidual organoids are odelling (glandular versus stroma/endometrial subtypes).

– Does the gestational age of TO and DO derivation correlate with the variability of expression of other cytokines? Or on any other readout?

– How was CM standardized across organoids (#, size, length after split, etc.)? Could hcg be used to standardize?

– For ease of interpretation, for Figure 6, please indicate what antigen is stained for directly under each image with words/colors as done in other figures.

– The current quantification of HCMV infection does not give a sense for the number of cells infected in each organoid. Consider an additional quantification that would better capture infection dynamics.

– Are these productive infections?

– Is the lack of cytokine and transcriptional response to HCMV infection simply due to the Tos not being infected? This should be included in discussion.

– Despite commonly indicating TO and DO are matched, the use of this matched specificity is not utilized either experimentally or in data interpretation. Does matched matter?

– Does TO CM prevent infection of DO? Does it act in paracrine to relevant cell types as concluded?

– Can TO CM prevent HCMV infection? This seems quite relevant given the focus on HCMV infection dynamics and the stark differences between Zika (an emerging RNA virus) and HCMV (DNA virus that has coevolved with host).

*Reviewer #2 (Recommendations for the authors):*

Listed below are my questions:

(1) The Tos shown in Figure 1C seem largely epithelial with few syncytium. Typically there are sporadically formed portions of STB (syncytiotrophoblast) that form internally. This could be due to their mid-late gestation source. I would like to see some staining for SDC1, hCG, etc. Something to indicate the location and abundance of spontaneously formed STB. I think this piece of data would be important to the trophoblast field to show similarities and differences of the structure of 2^nd^-3^rd^ trimester Tos compared to first which show abundant STB formation.

(2) The TO medium from your methods is very similar to Turco’s TOM medium, but you do state that nicotinomide was added for improved proliferation. That is reasonable, but I also noticed that 10% FBS is also added but there is no rationale for this addition to the medium.

Listed below are my critiques:

(1) As stated in my public review, I am concerned about the “attracting forces” of cell culture medias and in vitro culture generally. It is stated that first trimester timepoints are not as susceptible to HCMV infection as later stages and it is shown that your Tos are not susceptible to HCMV. The productions of IFN-lambda2 could be responsible for this protection, but it is not proven to me that these organoids, while derived from mid-late gestation, continue to resemble those timepoints in vitro and this requires transcriptional or maybe epigenetic comparisons to other first and second trimester placental datasets.

2) Also stated in the public review- I would really like for the authors to differentiate Tos into EVT organoids and repeat HCMV infections to prove that EVTs are or are not susceptible to HCMV infection. Please provide whole staining/confocal analysis OR infect with the fluorescent HCMVs, dissociate, and perform flow cytometry analysis to demonstrate susceptibility of EVT Tos to HCMV compared to Tos in TOM.

(3) An additional experiment that would really improve the impact of this paper is to attempt a co-culture model with both Tos and Dos together and repeat HCMV infection. Could Tos protect Dos from infection?

---

## [Author Response]

Essential revisions:(1) There is only a single readout for viral infection with quantification being % infected organoid (which could include organoids with only a single cell or hundreds of cells infected). A more fine-tuned quantification seems necessary given the conclusions of the manuscript.

Titration of HCMV infections is technically challenging in general, but even more so given the low levels of infection in many cell systems, including our organoid models. Nonetheless, to address this concern, we now include bulk RNASeq data mapping HCMV-associated reads to two genes important for HCMV replication in infected Tos and Dos (Figure 6E in the revised manuscript), which supports the low levels of replication in Tos.

(2) More convincing experimental data that EVTs contained in Tos are not susceptible should be included. The reviewer suggests: “The authors should differentiate their organoids into EVT organoids as described in Sheridan et al., Nature Protocols 2020 and try infection with their fluorophore-tagged hCMV viruses. Spontaneously formed pre-EVT within TOM-treated organoids are not very mature compared to EVTs within properly differentiated EVT organoids. They could try infection and whole mount image, OR dissociate the organoids and run a flow cytometry experiment to quantify the proportion of infected cells within their EVT organoids and TOM (cytotrophoblastic and syncytiotrophoblastic) organoids. "

To address this concern, we applied the EVT differentiation protocol in Sheridan et al. Because this protocol has not previously been applied to TOs from full-term tissue, we also include key data to support the success of this protocol. We now show that TOs isolated from full-term tissue respond to the EVT differentiation protocol as described in Sheridan et al., which correlated with an increase in HLA-G positive cells from ~18% to ~96% in undifferentiated versus differentiated conditions (Figure 1H in the revised manuscript). This differentiation also correlated with a significant increase in MMP-2, which is released at high levels from first trimester EVTs (Figure 1I in the revised manuscript).

Because we successfully applied the EVT differentiation to TOs, we compared the levels of HCMV infection between undifferentiated and differentiated TOs. This showed that TOs differentiated to contain >95% HLA-G positive cells remained highly resistant to HCMV infection (Figure 6F and 6G of the revised manuscript), supporting our conclusion that EVTs are not the primary targets of HCMV in TOs.

3) A weakness in this work is that there was no characterization of the resulting organoids proving that they indeed transcriptionally or epigenetically model a second-third trimester timepoint" or really any experimental comparison with first trimester to see how gestation age is reflected/modeled.

To address this concern, we used publicly available bulk RNASeq datasets generated by our own group and others to perform comparative expression analysis between TOs and placental tissue generated from the first or second trimesters, or from full-term tissue. Hierarchical clustering analysis using key hormones and other factors associated with trophoblast structure and function revealed that TOs were most closely related to tissue explants from the first trimester (Figure 2C of the revised manuscript).

Reviewer #1 (Recommendations for the authors):– The introduction should include a description of trophoblasts organoid orientation and what the decidual organoids are modeling (glandular versus stroma/endometrial subtypes).

Lines 86-87 of the original manuscript read as follows: “Similarly, decidua organoids isolated from uterine glands recapitulate the transcriptional profile of their tissue of origin and respond to hormonal cues..”. However, to make this clearer, we now include the following statement on lines 124-125 “…cultured progenitor cells isolated from chorionic villi or uterine glands of the decidua…”.

– Does the gestational age of TO and DO derivation correlate with the variability of expression of other cytokines? Or on any other readout?

There were no apparent differences in cytokine levels or other readouts of TOs used in this study, regardless of the gestational stage in which they were isolated. We have included individual datapoints from unique organoid preparation throughout the manuscript to demonstrate the variability that we observe between organoids lines, which is expected given that they are from human tissue. Of note, we now include a comparison of the transcriptional profiles of the TOs used in our study to placental tissue from the first and second trimesters, and from full-term tissue, and these studies indicated that TOs are most closely associated transcriptionally with first trimester tissue (Figure 2C of the revised manuscript).

– How was CM standardized across organoids (#, size, length after split, etc.)? Could hcg be used to standardize?

We have added a clearer explanation of the quality control measures we take for CM, which are based on hCG levels and overall organoid numbers and morphology. This description can be found on lines 809-815 of the revised manuscript.

– For ease of interpretation, for Figure 6, please indicate what antigen is stained for directly under each image with words/colors as done in other figures.

This has been added, as requested.

– The current quantification of HCMV infection does not give a sense for the number of cells infected in each organoid. Consider an additional quantification that would better capture infection dynamics.

We now show infection data as a percent of CMV+ cells over total nuclei taken from two dimensional images, and as now described on the methods.

– Are these productive infections?

It is unclear whether the reviewer refers to productive infections in DOs or TOs. In the case of DOs, the reporters of both AD159r and TB40E strains of HCMV are fused to transcripts associated with late replication of HCMV, suggesting that these are in fact productive infections. In the case of TOs, the lack of signal for either reporter, coupled with the low levels of production of key virally-encoded transcripts required for HCMV replication (Figure 6E in revised manuscript), suggests that the infection is not productive.

– Is the lack of cytokine and transcriptional response to HCMV infection simply due to the TOs not being infected? This should be included in discussion.

Despite the low level of overall productive infection, there are more transcripts induced in TOs infected with HCMV than in DOs (282 transcripts (AD169r strain) or 173 transcripts (TB40E strain) in TOs and 95 (AD169r strain) or 117 (TB40E strain) in DOs). Thus, the differences observed do not seem to result from differences in overall infection levels.

– Despite commonly indicating TO and DO are matched, the use of this matched specificity is not utilized either experimentally or in data interpretation. Does matched matter?

There does not appear to be a requirement for the use of matched organoid cultures. In fact, we have successfully generated co-cultures of unmatched TOs and DOs, which appears to work equivalently well regardless of whether they were derived from unmatched tissue (it should be noted that the co-cultures used in this study are from matched tissue). We have made this clearer in the text (lines 814-815).

– Does TO CM prevent infection of DO? Does it act in paracrine to relevant cell types as concluded?

In the revised manuscript, we show that CM isolated from TOs reduces ZIKV infection in DOs (Figure 4F, 4G in the revised manuscript).

– Can TO CM prevent HCMV infection? This seems quite relevant given the focus on HCMV infection dynamics and the stark differences between Zika (an emerging RNA virus) and HCMV (DNA virus that has coevolved with host).

As requested by reviewer 2, and to address this question, we now provide data from co-cultures of TOs and DOs, which shows that the presence of TOs confers protection of DOs from HCMV infection (Figure 8I and 8J of the revised manuscript).

Reviewer #2 (Recommendations for the authors):Listed below are my questions:(1) The TOs shown in Figure 1C seem largely epithelial with few syncytium. Typically there are sporadically formed portions of STB (syncytiotrophoblast) that form internally. This could be due to their mid-late gestation source. I would like to see some staining for SDC1, hCG, etc. Something to indicate the location and abundance of spontaneously formed STB. I think this piece of data would be important to the trophoblast field to show similarities and differences of the structure of 2nd-3rd trimester TOs compared to first which show abundant STB formation.

We now include SDC-1 immunostaining from TOs, which shows that there are large numbers of SDC-1 positive cells in TOs (Figure 1C of the revised manuscript). We also note that TOs express high levels of many STB markers (e.g., CGBs, PSGs) as assessed by bulk RNASeq (Figure 2B and 2C of the revised manuscript).

(2) The TO medium from your methods is very similar to Turco's TOM medium, but you do state that nicotinomide was added for improved proliferation. That is reasonable, but I also noticed that 10% FBS is also added but there is no rationale for this addition to the medium.

Our optimization revealed that 10% enhanced the growth rate of TOs. This has been added to the methods.

Listed below are my critiques:(1) As stated in my public review, I am concerned about the "attracting forces" of cell culture medias and in vitro culture generally. It is stated that first trimester timepoints are not as susceptible to HCMV infection as later stages and it is shown that your TOs are not susceptible to HCMV. The productions of IFN-lambda2 could be responsible for this protection, but it is not proven to me that these organoids, while derived from mid-late gestation, continue to resemble those timepoints in vitro and this requires transcriptional or maybe epigenetic comparisons to other first and second trimester placental datasets.

In the revised manuscript, we show that the transcriptional signature of TOs more closely resembles placental tissue isolated from the first trimester than from the second trimester of from full-term tissue. Other groups have also demonstrated the remarkable resistance of first trimester placental tissue to viral infections (e.g., PMID: 27443522).

(2) Also stated in the public review- I would really like for the authors to differentiate TOs into EVT organoids and repeat HCMV infections to prove that EVTs are or are not susceptible to HCMV infection. Please provide whole staining/confocal analysis OR infect with the fluorescent HCMVs, dissociate, and perform flow cytometry analysis to demonstrate susceptibility of EVT TOs to HCMV compared to TOs in TOM.

As indicated above, we have performed EVT differentiation, which we fully characterized (Figure 1H and II in the revised manuscript) and show that this does not impact the levels of HCMV infections (Figure 6F and 6G of the revised manuscript).

(3) An additional experiment that would really improve the impact of this paper is to attempt a co-culture model with both TOs and DOs together and repeat HCMV infection. Could TOs protect DOs from infection?

As suggested, we established co-cultures of TOs and DOs and found that presence of TOs in these co-cultures significantly decreased the susceptibility of DOs to HCMV infection (Figure 8I and 8J in the revised manuscript)